# The relative effects of self-reported noise and odour annoyance on psychological distress: Different effects across sociodemographic groups?

Eline Berkers[1,2]*, Ioana Pop[1], Mariëlle Cloïn[2‡], Antje Eugster[2,3‡], Hans van Oers[2,4‡]

1 Department of Sociology, Tilburg University, Tilburg, The Netherlands, 2 Tranzo, Tilburg University, Tilburg, The Netherlands, 3 GGD Brabant-Zuidoost, Eindhoven, The Netherlands, 4 Ministry of Health, Welfare and Sports, The Hague, The Netherlands

☯ These authors contributed equally to this work.
‡ These authors also contributed equally to this work.
* e.c.m.berkers@tilburguniversity.edu

## Abstract

In earlier research, both higher levels of noise and odour annoyance have been associated with decreased mental health. Presumably, these perceptions can trigger feelings of threat and stress reactions and in turn evoke psychological distress. There are two important lacunas in the research on this topic: most studies only consider either noise or odour annoyance and not their relative effect on psychological distress and there is scarce evidence about whether different sociodemographic groups experience more psychological distress due to noise and odour annoyance. Starting from the diversity in the available coping resources and in their daily life patterns, we distinguish gender, age and educational level as relevant sociodemographic variables. Using data from the Health Monitor (n = 25236) in Noord-Brabant, we found using Ordinary Least Squares Regression that individuals that reported higher levels of noise and odour annoyance reported higher levels of psychological distress. Furthermore, the effect of noise annoyance was relatively stronger compared to that of odour annoyance. Regarding the interaction effects, we found that younger adults' psychological distress was more strongly affected by noise annoyance compared to older adults, but not by odour annoyance. The psychological distress of individuals with no or primary education was more strongly affected by both noise and odour annoyance compared those with tertiary education, but not when compared to those who completed lower or higher secondary education. Contrary to our expectations, we did not find different effects between men and women. Though the evidence for the interactions was mixed, classic health inequalities along age and education lines are reinforced when considering the relationship between noise and odour annoyance and psychological distress.

**Data Availability Statement:** The data cannot be shared due to legal restrictions. The data are owned by a third-party organization, i.e. the three

local regional health service organizations (GGD Brabant-Zuidoost, GGD Hart voor Brabant & GGD West-Brabant) that serve as partner organizations in this research. The authors were given permission to use the data for this research project, but are not legally allowed to share this data outside the project. All data that is used in this study are available from the website of the GGD GHOR (the overarching organization to which the three partner organizations belong) if researchers meet the criteria. Data requests can be done at: https://monitorgezondheid.nl/contact or by email to monitorgezondheid@ggdghor.nl.

**Funding:** The authors received no specific funding for this work.

**Competing interests:** The authors have declared that no competing interests exist.

## Introduction

Over the years, the mental health disadvantages of living in an unruly environment filled with environmental stressors have become clear [1–5]. Two of these stressors that negatively affect mental health are noise and odour, e.g., the World Health Organization Regional Office for Europe [6] estimates that every year more than 1 million healthy years are lost due to traffic noise only. In addition, odour pollution is associated with many health complaints and decreased quality of life [7]. If we add to this the fact that mental disorders, and especially mood disorders, pose a very high burden on the population of Europe and are expected to become even more prominent for population health in the future [8, 9], it is clear that a better understanding of the relationship between odour and noise and mental health is needed in order to mitigate these noxious pathways.

Noise and odour can be quantified by using decibels and odour standards (for example, ammonia thresholds) and are thought to affect individual's mental health in two different ways: a direct pathway, i.e., a negative effect on the nervous system through physical arousal and an activation of stress hormones, and an indirect pathway through triggering negative emotional reactions such as anger or stress [10–12]. However, people are differently affected by noise or odour exposure: what is annoying to some may not be to others depending on personal, social or general characteristics, such as personality, sensitivity to noise or odour, relationship with the source, ability to 'escape' and history of disturbance due to noise or odour [13–15]. Subsequently, the relationship between noise exposure and noise annoyance is not that clear-cut and only around ⅓ of the variance in noise annoyance is explained by exposure to noise [13]. Similar sources are not available for odour but it is reasonable to assume a similar association.

In general, subjective perceptions of noise and odour more closely reflect the actual impact of daily life environmental circumstances than more objective ones simply because individuals are not fully aware of the objective circumstances in their living environment [16] given that they rely on their senses for perceiving them [17]. Furthermore, subjective perceptions are the pathway between objective stressors and the resulting emotional states (i.e., psychological distress) [18, 19]. Moreover, noise and odour form the top two environmental complaints for European citizens and therefore is an important issue for policy [7]. For policy makers at a municipality or province, sophisticated exposure data is hard to interpret and translate into policy. Therefore, studying the relationship between noise and odour annoyance and psychological distress is important, both from a scientific and policy perspective. Subsequently, in this contribution we focus on the noise and odour annoyance experienced by individuals and their impact on psychological distress.

Previous evidence from cross-sectional studies relates noise annoyance stemming from different sources (e.g. industry, traffic or neighbours) to increased levels of psychological distress among adults, both in urban and rural areas, and in different countries [18, 20–25]. Furthermore, a recent longitudinal study by Beutel, Brähler [26] found that even after controlling for current levels of noise annoyance, the experience of past noise annoyance, as far as five years ago, was linked to a higher probability of anxiety and depressive symptoms. Turning to odour annoyance, previous studies report that higher levels of odour annoyance due to agricultural activities are associated with higher levels of psychological distress among adults living in rural areas [11, 27–29].

This said, our first contribution to the literature is our focus on the relationship between both noise and odour annoyance and psychological distress as opposed to focusing on each factor separately. In this way, we account for the fact that noise and odour annoyance can occur at the same time (i.e., they can originate from the same source, e.g., industrial activity or

traffic). By taking into account both noise and odour annoyance, this study will provide more insight into their relative contribution to psychological distress [23].

Second, the present study responds to calls in the literature to more closely examine *for whom* the daily living conditions (e.g., noise and odour annoyance) have a stronger effect on psychological distress [30, 31]. Life is full of low-impact factors (daily hassles, like noise and odour annoyance) that add up and use the coping resources available for people and in this way have a substantial impact on individuals [19]. Different sociodemographic groups have access to different coping resources and have different daily living patterns, which influence their capacity to cope with environmental stressors. As a result, certain groups might experience higher levels of psychological distress due to perceived noise and odour annoyance. Answering the question whether different groups are impacted differently by these stressors helps to understand the causes of health inequalities across different groups. However, so far, the scientific evidence supporting this assumption has been scarce, as we will detail later in the theoretical section.

To sum up, the research questions in this study are: how are noise and odour annoyance related to psychological distress? What is the relative contribution of noise and odour annoyance to psychological distress? Is this relationship different for groups defined by gender, age and educational level? We address these questions based on data of adult respondents (n = 25236) collected in a province in the South of the Netherlands (Noord-Brabant). In this part of the Netherlands, livestock farming and industry are important sources of nuisance [32] and both the population density and livestock density in this province rank very high [33, 34]. Next, the province is home to one commercial and a few military airports. The variation of these contextual stressors between different living areas makes this an adequate area to study this subject.

## Relationship between noise and odour annoyance and psychological distress

To understand why noise and odour annoyance could increase psychological distress, we draw from two related theoretical models, i.e., the transactional stress model proposed by Lazarus and Folkman [19] and the Conservation of Resources theory [35, 36]. Lazarus and Folkman [19] proposed that the subjective perception of a contextual stressor determines the actual strength of the individual stress reaction. Implicitly, individuals rely on their senses to appraise the conditions in their living environment [17]. If individuals appraise their living conditions as harmful to their wellbeing now or in the future, this creates a disbalance between the individual and its living environment. This in turn can lead to psychological distress [19, 37]. For the case of noise and odour, potential health risks are an increased probability of developing concentration or sleeping issues (for noise) or irritation on the eyes, nose and throat or respiratory issues (for odour) [27, 38, 39]. Subsequently, the appraisal of these potential risks in the environment can lead to feelings of anger, anxiety and stress, or in more general terms, to psychological distress.

The description of this process details how individuals' perception of an acute stressor influences their levels of psychological distress. When the stressor is environmental, individuals are often confronted with it over longer periods of time, because individuals have relatively stable living arrangements [40]. These repeated confrontations with noise and odour annoyance will undermine the basic need of individuals to live in a clean and predictable living environment and instead paint a picture of a living environment that is chaotic and full of threats to individuals' health [4].

According to the Conservation of Resources theory, in order to deal with these stressful conditions, individuals need to activate their coping resources, such as socioeconomic and

health resources [4, 35, 41]. Coping takes effort (e.g., by taking political action or minimizing exposure by closing windows or doors) and if individuals have to persistently use resources to deal with noise and odour annoyance, they are at a greater risk of diminished ability to deal with future stressors [36, 42]. In addition, if their coping effort is unsuccessful, individuals are at risk to lose further coping resources [35]. This mechanism implies that individuals that are confronted with noise and odour annoyance will report higher levels of psychological distress.

Moreover, noise and odour annoyance can lead to psychological distress through the persistent disturbance of daily activities, such as relaxation (e.g., reading or watching television) or restoration (e.g., sleeping) which are essential for replenishing cognitive and physical resources that are used throughout the day [43]. Individuals that are unable to replenish these resources will have more difficulties in coping with stressors, which will lead to increased psychological distress [44].

Based on the above arguments we expect that individuals that report higher levels of noise annoyance (hypothesis 1a) and odour annoyance (hypothesis 1b) report higher levels of psychological distress.

Secondly, regarding the relative impact of noise and odour annoyance on psychological distress, we expect that noise annoyance will have a stronger relative impact. For odour annoyance there is more variation in the levels of annoyance in comparison to noise as a result of seasonal influences [23]. Noise, while depending to some extent on the wind direction, has a more constant presence in the environment, which results in a relatively larger burden for psychological distress compared to odour annoyance. Thus, our hypothesis is that noise annoyance will have a stronger effect on psychological distress compared to odour annoyance (hypothesis 2).

## Different relationships across sociodemographic groups

As noted in the introduction, the differential effect of noise and odour annoyance on psychological distress for different sociodemographic groups is scarce [45]. However, the same process of coping can be used as a starting point to elaborate on these effects. Not surprising, the classic social inequality markers of gender, age and education are undisputable main dimensions of inequalities in coping resources. If individuals rely on coping resources in order to mitigate the effect of noise and odour annoyance on psychological distress, social groups with higher levels of such resources will fare better.

We identify personal control, financial resources and cognitive or physical health capacities as relevant buffers in between noise and odour annoyance and psychological distress. First, a lack of personal control, defined as 'the belief that one can master, control and shape one's own life' [46] increases the perceived threat posed by the environmental stressors because individuals feel unable to avoid their negative consequences [5, 46]. As a result, individuals with lower levels of personal control, i.e. women, older adults and less educated [4, 47] are unlikely to act to change their situation and they develop negative emotions [48].

Second, individuals with more financial resources can better protect themselves from the negative effects of noise and odour annoyance because they can afford better housing (with better isolation) and are more likely to have political influence to prevent negative environmental changes around their home [41]. In general, women and older adults report lower levels of financial resources, e.g., they make fewer working hours due to part-time employment or retirement [49, 50]. Furthermore, a higher education level increases the opportunities for good jobs with a higher income [47].

And third, the experience of decreasing cognitive and physical health already puts individuals at a risk for increased psychological distress [35], adding to the negative effects of the

precarious environmental conditions. Furthermore, individuals experiencing cognitive or physical health decline engage their financial and social resources, as well as their time and energy, to cope with their health situation [35], leaving less resources available to deal with noise and odour annoyance. The later middle-age and the older age are the phases where physical and cognitive health issues are more likely to arise, thus this argument could mostly apply to these age groups [51].

Next to differences in coping resources, sociodemographic groups defined by gender, age and education also have different daily living patterns [52], which can contribute to the differential susceptibility of being affected by noise and odour annoyance. One particular difference is the amount of time spent at home. This is an important indicator of noise and odour annoyance [53, 54]. It is reasonable to expect that the more people are exposed to noise and odour annoyance in an environment that they cannot easily escape, such as their own homes, the more likely it is that this annoyance will increase the psychological distress because it undermines the basic need for a peaceful living environment [44]. Women, older adults and less educated spend more time at home than men, younger adults and higher educated [55].

To sum up, based on the difference in coping resources and daily living patterns between the three sociodemographic groups, we can derive the following hypotheses. First, we expect the psychological distress of women to be affected more strongly by noise annoyance (hypothesis 3a) and odour annoyance (hypothesis 3b) than men. Next, we expect that older adults' psychological distress is affected more strongly by noise annoyance (hypothesis 4a) and odour annoyance (hypothesis 4b) compared to middle-aged and younger adults. Finally, individuals with lower levels of education will experience less psychological distress when confronted with noise annoyance (hypothesis 5a) and odour annoyance (hypothesis 5b) compared to individuals with higher levels of education.

## Data and method

We used data from the Health Monitor collected by the Dutch Regional Health Services (GGD), Statistics Netherlands (CBS) and the National Institute for Public Health and the Environment (RIVM), collected between September and December in 2016. The data is collected every four years with the purpose of gathering relevant information about the public health situation in the Netherlands on different spatial levels (e.g., regional, national, local). In our main analysis, we use a sample that consists of adults (aged 19–64, n = 29647) from the province of Noord-Brabant (divided in three subregions: West, Middle and South-East Brabant). We replicate the main models on a larger sample which includes older adults aged 65+ (n = 34838, of which 25326 adults and 9512 older adults) because we expect older adults' psychological distress to be more strongly affected by noise and odour annoyance. This sample was not used in the main analyses because it was drawn from the same subregion (South-East Brabant) and the respondents answered a subset of the items in the noise annoyance scales (three out of nine). Approval was obtained for the publication of the content by the Ethics Review Board of the school of Social and Behavioral Sciences at Tilburg University.

The sampling was done by Statistics Netherlands based on the Municipal Personal Records Database. Respondents were invited to participate in a web survey, which included topics such as health, lifestyle, perceived neighbourhood quality and social contacts (for more specific information about the questionnaire: visit the website of the National Institute for Public Health and the Environment [56]). Respondents were offered a paper questionnaire as an alternative to the web survey. In our sample, 74.4% of the adults filled in the web survey and the rest the paper survey. The overall response rate for Noord-Brabant was 32% for adults, which is slightly lower than the overall response rate for the entire survey (40%). After deleting

the cases with missing values (n = 4411, about 15% of the full sample) on the main variables, the final included sample of adult respondents was 25236. Using Little's MCAR test, we found that the missing values in our analysis were not completely missing at random (as the p-value was significant at an alpha of 0.05). To evaluate if there were differences in the missing values on psychological distress across the different sociodemographic groups, we used a cross tabulation to calculate the chi-square statistic. There were no significant differences in missing values among men and women, but those with lower levels of education and younger individuals were more likely to have missing values compared to those with higher educational levels and older individuals. We did not consider imputation methods for our missing values, because this is an estimation method which depends on the assumptions of the model that is estimated. Since we had a reasonable amount of missing values, we chose to rely on listwise deletion.

In our sample, 54.5 percent was female and the majority of the sample was aged between 45 and 64 (48.7 percent), whereas younger adults were slightly underrepresented (21.1 percent) compared to the other age groups.

## Psychological distress

The survey included the Kessler-10 scale, which is commonly used to measure nonspecific psychological distress (e.g., symptoms of depression and anxiety disorders) [57, 58]. The participant was asked to think about how he or she felt in the past 4 weeks. Examples of questions are: 'how often did you feel tired without a reason?', 'how often did you feel hopeless?' or 'how often did you feel so restless that you could not sit still?' Respondents could answer between 1) 'never' and 5) 'all of the time'. All items loaded on one factor and the scale had a Cronbach's alpha of 0.91. Respondents that did not answer all questions were filtered out of the data (n = 3421). We computed a sum score, but to ease the interpretation of the results we rescaled the scale to range between 0 and 100, where a higher score represented higher levels of psychological distress. Subsequently, we can interpret the coefficients from our regression models as changes in percentage points. As part of our sensitivity analyses, we re-estimated the main models using a dichotomous dependent variable. We followed the recommendation by Donker, Comijs [57] and we used a cut-off point of 25 out of 100 to differentiate between participants at risk and those not at risk of clinical depression and anxiety disorder.

## Noise annoyance

Respondents indicated to what extent they were hindered or annoyed by noise from different sources in the last twelve months: a) roads below < 50 km, b) roads above > 50 km, c) railways, d) aircraft, e) scooters, f) neighbours, g) industry or companies, h) (re)construction sites or, i) restaurants, cafes or bars. The measurement scale ranged from 0 (not annoyed) to 10 (highly annoyed). The items loaded on one factor (Cronbach's alpha = 0.76). The survey included a separate category for individuals that could not hear the noise at home. Respondents that gave this answer were given a score of 0 as if they indicated being able to hear the noise and were not disturbed by it. Like previous studies that took into account multiple sources of noise [22, 59], a mean score was computed based on a minimum of eight out of the nine items (n = 352 individuals answered only eight items).

For our analyses, we used the noise annoyance scale as a continuous variable ranging from 0 "no odour annoyance" to 10 "high odour annoyance" (we present the distribution of scores in Fig 1 in the S1 Appendix). For the older adult sample, only three out of the nine items were available to compute the average noise annoyance score, namely noise annoyance due to industry or companies, (re)construction sites or restaurants, cafes and bars.

## Odour annoyance

We measured odour annoyance using an item that asked the respondents if they experienced annoyance or hinder during the last twelve months in their home due to: a) roads, b) sewerage, c) fireplaces, d) agriculture, e) industry, f) stables, g) manure, h) animal feed, i) a digester or j) aircraft traffic. Answers ranged from 0 (not annoyed) to 10 (highly annoyed). Again, respondents that indicated not being able to smell the odour at home were scored 0 as if they were not annoyed by this odour, but could smell it. We performed an exploratory factor analysis that revealed two factors: one related to agriculture (items d, f, g and h) and one relating to other types of odour (items a, b, c, e, i and j). In total, 62% of the respondents indicated that they were not annoyed by odour by agriculture and 45% indicated the same referring to odour annoyance due to other sources. This makes sense given that agricultural odour annoyance will be more present in rural areas, whereas odour annoyance due to other sources will be omnipresent (e.g., due to fireplaces, roads, sewerage). We computed a mean score on a minimum of three out of four items for agricultural odour (n = 123 respondents with 3 items) and five out of six items for other odour annoyance (n = 234 respondents with 5 items). Both scales had good reliability, i.e., Cronbach's alpha was 0.86 for agricultural odour annoyance scale and 0.67 for the scale measuring annoyance due to other sources. We used the two scales as continuous variables that ranged from 0 "no odour annoyance" to 10 "high odour annoyance" (information about the distribution of scores can be found in Figs 2 and 3 in the S1 Appendix).

## Sociodemographic groups

Following the terminology by Pai and Kim [51] and Lachman [50] we divided the sample into three age groups: age between 19 and 35 (younger adults, ref.), between 36 and 50 (young middle-aged) and 51 and 64 (older middle-aged). In the analysis with the older adult sample, those older than 64 form a separate age group. For gender, men formed the reference category. For educational level, four educational levels were taken into account: no or primary (ref.), lower secondary, higher secondary and tertiary education.

## Control variables

Given that individuals with a lower socioeconomic status are more exposed to noise because of their living situation (e.g., worse housing conditions) and are more at risk for psychological distress, we controlled for subjective economic wellbeing [39, 59]. For subjective economic wellbeing, respondents indicated whether they had experienced financial hardship in the last twelve months or not (ref. is no). In addition, we controlled for self-rated health, because on the one hand, worse health increases the vulnerability for environmental stressors [15] and on the other hand, is strongly related to mental health [18, 22, 60]. Self-rated health was measured by the following question: how do you perceive your health in general? [61]. Three answer options were used: bad, moderate and good health (ref.). We also took into account parenthood (measured by cohabitation with a child under 18, ref. is no). Parents are more likely to spend time around their home environment due to child-rearing duties [62] and children form an extra reason for concern about noise and odour annoyance [63]. Next, parents' child-rearing duties and work-life balance take up time and energy and consequently lead to higher levels of psychological distress [55, 64, 65].

Because we do not have access (for data protection reasons) to detailed address data, we cannot model actual noise and odour exposure for respondents. Alternatively, to get an indication of noise and odour exposure around the home, we modelled indicator variables for all districts and self-reported noise and odour annoyance around the home. Following the study by van Deurzen, Rod [40], we used indicator variables to control for differences in the objective

living conditions and exposure to noise and odour between the 306 districts in the sample (size between 42 and 272 respondents). The first in the list (Wijk en Aalburg) is used as the reference category. Finally, we controlled for the self-reported risks of noise and odour annoyance in the vicinity of the home (e.g., a busy street, wind turbine, agriculture or a gas station) by taking into account whether they are present or not (ref. is no).

Descriptive statistics for the variables in our analyses are summarized in Table 1.

## Analytical strategy

In this study, we relied on bivariate analysis of mean scores and ordinary least squares regressions (as our dependent variable is a scale) using Stata 16.0. For our bivariate analysis, we showed the average psychological distress scores of those with different scores on the noise and odour annoyance scale to illustrate if there are differences between these groups.

**Table 1. Descriptive statistics of all variables (n = 25236).**

|  | Min | Max | Mean / % | SD |
|---|---|---|---|---|
| Psychological distress | 0 | 100 | 15.65 | 15.68 |
| Noise annoyance | 0 | 10 | 1.29 | 1.32 |
| Odour annoyance–agriculture | 0 | 10 | 0.82 | 1.58 |
| Odour annoyance–other | 0 | 10 | 0.70 | 1.08 |
| Female |  |  | 54.5% |  |
| *Age category* |  |  |  |  |
| Young adults |  |  | 21.1% |  |
| Young middle-aged |  |  | 30.3% |  |
| Older middle-aged |  |  | 48.7% |  |
| *Educational level* |  |  |  |  |
| No or primary education |  |  | 2.2% |  |
| Lower secondary |  |  | 21.7% |  |
| Higher secondary |  |  | 39.2% |  |
| Tertiary |  |  | 36.9% |  |
| *Control variables* |  |  |  |  |
| Lives with children under 18 |  |  | 30.7% |  |
| Self-rated health |  |  |  |  |
| (Very) Good |  |  | 76.6% |  |
| Moderate |  |  | 17.3% |  |
| (Very) Bad |  |  | 3.1% |  |
| *Subjective financial wellbeing* |  |  |  |  |
| Yes vs. no |  |  | 14.1% |  |
| *Self-reported risks in the environment* |  |  |  |  |
| Lives in a busy street |  |  | 22.7% |  |
| Lives near industry |  |  | 16.5% |  |
| Living near airport |  |  | 11.6% |  |
| Living near a livestock farm |  |  | 25.4% |  |
| Living near a wind turbine |  |  | 2.8% |  |
| Living near agriculture (without livestock) |  |  | 30.9% |  |
| Living near a route for dangerous materials |  |  | 8.6% |  |
| Living near a gas station |  |  | 16.6% |  |

Source: Health Monitor (2016).

**Table 2. Selected effects of OLS regression for psychological distress.**

| | Model 1 | | | Model 2 | | | Model 3 | | |
|---|---|---|---|---|---|---|---|---|---|
| | B (se) | Beta | P-value | B (se) | Beta | P-value | B (se) | Beta | P-value |
| Noise annoyance | 1.86 (0.07) | 0.16 | 0.00 | - | - | - | 1.53 (0.09) | 0.13 | 0.00 |
| Odour annoyance–agriculture | - | - | - | 0.41 (0.08) | 0.04 | 0.00 | 0.23 (0.08) | 0.02 | 0.00 |
| Odour annoyance–other | - | - | - | 1.36 (0.11) | 0.09 | 0.00 | 0.45 (0.12) | 0.03 | 0.01 |
| *Age category (ref. young adults)* | | | | | | | | | |
| Young middle-aged | -3.07 (0.27) | -0.09 | 0.00 | -3.08 (0.27) | -0.09 | 0.00 | -3.12 (0.27) | -0.09 | 0.00 |
| Older middle-aged | -4.35 (0.25) | -0.14 | 0.00 | -4.48 (0.26) | -0.14 | 0.00 | -4.47 (0.25) | -.14 | 0.00 |
| Female | 2.33 (0.19) | 0.07 | 0.00 | 2.31 (0.19) | 0.07 | 0.00 | 2.32 (0.19) | 0.07 | 0.00 |
| *Educational level (ref. tertiary)* | | | | | | | | | |
| No / primary | 10.45 (0.64) | 0.10 | 0.00 | 9.85 (0.64) | 0.09 | 0.00 | 10.30 (0.64) | 0.10 | 0.00 |
| Lower secondary | 3.62 (0.27) | 0.10 | 0.00 | 3.20 (0.27) | 0.08 | 0.00 | 3.56 (0.27) | 0.09 | 0.00 |
| Higher secondary | 1.47 (0.22) | 0.05 | 0.00 | 1.21 (0.22) | 0.04 | 0.00 | 1.43 (0.22) | 0.04 | 0.00 |
| *Control variables* | | | | | | | | | |
| Parenthood | - | - | - | - | - | - | - | - | - |
| *Self-rated health (ref. good)* | | | | | | | | | |
| Moderate | - | - | - | - | - | - | - | - | - |
| Bad | - | - | - | - | - | - | - | - | - |
| *Lives near:* | | | | | | | | | |
| A busy street | -0.34 (0.23) | -0.01 | 0.14 | 0.33 (0.23) | 0.01 | 0.16 | -0.32 (0.23) | -0.01 | 0.17 |
| Industry | -0.28 (0.27) | -0.01 | 0.31 | -0.05 (0.27) | -0.00 | 0.85 | -0.30 (0.27) | -0.01 | 0.27 |
| Airport | -0.59 (0.39) | -0.01 | 0.13 | -0.27 (0.39) | -0.01 | 0.50 | -0.65 (0.39) | -0.01 | 0.10 |
| Livestock farm | 0.37 (0.30) | 0.01 | 0.22 | -0.01 (0.31) | -0.00 | 0.96 | 0.14 (0.31) | -0.00 | 0.65 |
| Wind turbine | 0.33 (0.63) | 0.00 | 0.60 | 0.38 (0.63) | 0.00 | 0.54 | 0.32 (0.63) | 0.00 | 0.61 |
| Agriculture (without livestock) | -0.07 (0.28) | -0.00 | 0.81 | -0.20 (0.29) | -0.01 | 0.49 | -0.13 (0.28) | -0.00 | 0.65 |
| A route for dangerous materials | -0.27 (0.35) | -0.00 | 0.44 | 0.17 (0.35) | 0.00 | 0.62 | -0.30 (0.35) | -0.01 | 0.39 |
| Gas station | 0.25 (0.27) | 0.01 | 0.34 | 0.38 (0.27) | 0.01 | 0.16 | 0.26 (0.27) | 0.01 | 0.33 |
| Intercept | 9.40 (1.92) | | 0.00 | 10.75 (1.92) | | 0.00 | 9.60 (1.91) | | 0.00 |
| *Explained variance* | 14.8 | | | 14.0 | | | 14.9 | | |

Notes: all models are controlled for subjective financial wellbeing and indicator variables for all districts (ref. Wijk en Aalburg).

Source: Health Monitor (2016).

Moreover, the results of the regression models are presented in Tables 2 and 3. In Table 2, Model 1 we estimated the effect of the noise annoyance measure, controlling for the sociodemographic and environmental control variables. Model 2 included both scales of odour annoyance and Model 3 both noise annoyance and odour annoyance measures (controlling only for the sociodemographic and environmental control variables). In Table 3, Model 1 we added parenthood as a control variable. In Model 2, we replaced the parenthood measure with self-rated health and in Model 3 we included both parenthood and self-rated health. All interaction effects are based on Model 3. In Tables 4 and 5 we presented the significant effects of the estimated interactions of noise and odour annoyance measures with gender, age groups and educational levels. For comparison purposes of the strength of the effect, besides the unstandardized coefficients we also present the standardized ones (beta). We also present the analyses for the sample that includes the older adults in the S1 Appendix. As a sensitivity analysis, we estimated our models for a dependent variable that differentiated between individuals with and without a risk of clinical depression and anxiety disorder by using logistic regression.

**Table 3. Selected effects of OLS regression for psychological distress.**

| | Model 1 | | | Model 2 | | | Model 3 | | |
|---|---|---|---|---|---|---|---|---|---|
| | B (se) | Beta | P-value | B (se) | Beta | P-value | B (se) | Beta | P-value |
| Noise annoyance | 1.52 (0.09) | 0.13 | 0.00 | 1.33 (0.08) | 0.11 | 0.00 | 1.33 (0.08) | 0.11 | 0.00 |
| Odour annoyance–agriculture | 0.22 (0.08) | 0.02 | 0.00 | 0.22 (0.07) | 0.02 | 0.00 | 0.22 (0.07) | 0.02 | 0.00 |
| Odour annoyance–other | 0.47 (0.12) | 0.03 | 0.00 | 0.30 (0.11) | 0.02 | 0.01 | 0.31 (0.11) | 0.02 | 0.01 |
| *Age category (ref. young adults)* | | | | | | | | | |
| Young middle-aged | -2.20 (0.28) | -0.06 | 0.00 | -3.62 (0.24) | -0.11 | 0.00 | -3.00 (0.26) | -0.09 | 0.00 |
| Older middle-aged | -4.96 (0.26) | -0.16 | 0.00 | -5.95 (0.23) | -0.19 | 0.00 | -6.26 (0.24) | -.0.20 | 0.00 |
| Female | 2.34 (0.19) | 0.07 | 0.00 | 2.24 (0.17) | 0.07 | 0.00 | 2.25 (0.17) | 0.07 | 0.00 |
| *Educational level (ref. tertiary)* | | | | | | | | | |
| No / primary | 10.02 (0.64) | 0.10 | 0.00 | 5.77 (0.59) | 0.05 | 0.00 | 5.61 (0.59) | 0.05 | 0.00 |
| Lower secondary | 3.28 (0.27) | 0.09 | 0.00 | 1.95 (0.25) | 0.05 | 0.00 | 1.77 (0.25) | 0.05 | 0.00 |
| Higher secondary | 1.27 (0.22) | 0.04 | 0.00 | 0.78 (0.20) | 0.02 | 0.00 | 0.68 (0.20) | 0.02 | 0.00 |
| *Control variables* | | | | | | | | | |
| Parenthood | -2.52 (0.24) | -0.07 | 0.00 | - | - | - | -1.68 (0.22) | -0.05 | 0.00 |
| *Self-rated health (ref. good)* | | | | | | | | | |
| Moderate | - | - | - | 12.44 (0.23) | 0.30 | 0.00 | 12.36 (0.23) | 0.30 | 0.00 |
| Bad | - | - | - | 25.41 (0.50) | 0.28 | 0.00 | 25.28 (0.50) | 0.28 | 0.00 |
| *Lives near:* | | | | | | | | | |
| A busy street | -0.32 (0.23) | -0.01 | 0.17 | -0.29 (0.21) | -0.01 | 0.17 | -0.29 (0.21) | -0.01 | 0.17 |
| Industry | -0.31 (0.27) | -0.01 | 0.25 | -0.20 (0.25) | -0.01 | 0.43 | -0.20 (0.25) | -0.01 | 0.42 |
| Airport | -0.66 (0.39) | -0.01 | 0.09 | -0.51 (0.36) | -0.01 | 0.16 | -0.52 (0.36) | -0.01 | 0.15 |
| Livestock farm | 0.15 (0.31) | 0.00 | 0.63 | 0.04 (0.28) | 0.00 | 0.90 | 0.04 (0.28) | 0.00 | 0.88 |
| Wind turbine | 0.34 (0.63) | 0.00 | 0.59 | 0.42 (0.57) | 0.00 | 0.46 | 0.43 (0.57) | 0.01 | 0.45 |
| Agriculture (without livestock) | -0.09 (0.28) | -0.00 | 0.75 | 0.07 (0.26) | 0.00 | 0.80 | 0.09 (0.26) | 0.00 | 0.72 |
| A route for dangerous materials | -0.31 (0.35) | -0.01 | 0.37 | -0.12 (0.32) | -0.00 | 0.70 | -0.13 (0.32) | -0.00 | 0.68 |
| Gas station | 0.18 (0.27) | 0.00 | 0.50 | -0.04 (0.24) | -0.00 | 0.86 | -0.09 (0.24) | -0.00 | 0.71 |
| Intercept | 10.62 (1.91) | | 0.00 | 9.20 (1.75) | | 0.00 | 9.88 (1.75) | | 0.00 |
| *Explained variance* | 15.3 | | | 29.0 | | | 29.1 | | |

Notes: all models are controlled for subjective financial wellbeing and indicator variables for all districts (ref. Wijk en Aalburg).

Source: Health Monitor (2016).

# Results

Simple bivariate analyses (Table A1 in the S1 Appendix) showed that those who were least annoyed by noise had the lowest average psychological distress score (score ranges from 0 to

**Table 4. Interaction effects of noise annoyance for different age groups.**

| | B (SE) | Beta | P-value |
|---|---|---|---|
| Noise annoyance | 1.57 (0.16) | 0.13 | 0.00 |
| *Interaction with (ref. young adults)* | | | |
| Young middle-aged | -0.13 (0.19) | -0.01 | 0.51 |
| Older middle-aged | -0.41 (0.17) | -0.03 | 0.02 |

Notes: all models are controlled for subjective financial wellbeing, educational level, self-rated health, self-reported risks in the environment and indicator variables for all districts (ref. Wijk en Aalburg).

Source: Health Monitor (2016).

**Table 5. Interaction effects of noise and odour annoyance for educational level.**

| | Noise annoyance | | | Odour annoyance agriculture | | | Odour annoyance other | | |
|---|---|---|---|---|---|---|---|---|---|
| | B (SE) | Beta | P-value | B (SE) | Beta | P-value | B (SE) | Beta | P-value |
| Direct effect | 1.22 (0.12) | 0.10 | 0.00 | 0.16 (0.10) | 0.02 | 0.12 | 0.18 (0.16) | 0.01 | 0.26 |
| *Interaction with educational level (ref. tertiary)* | | | | | | | | | |
| No / primary | 1.33 (0.37) | 0.03 | 0.00 | 0.81 (0.32) | 0.02 | 0.01 | 1.63 (0.39) | 0.03 | 0.00 |
| Lower secondary | 0.20 (0.17) | 0.01 | 0.25 | 0.18 (0.14) | 0.01 | 0.22 | 0.22 (0.21) | 0.01 | 0.30 |
| Higher secondary | 0.08 (0.15) | 0.01 | 0.58 | -0.00 (0.12) | -0.00 | 0.99 | 0.00 (0.19) | 0.00 | 0.99 |

Notes: all models are controlled for subjective financial wellbeing, educational level, self-rated health, self-reported risks in the environment and indicator variables for all districts (ref. Wijk en Aalburg).

Source: Health Monitor (2016).

under 3; 14.9) as compared to those who were annoyed (score ranges from 3 to under 8; 21.1) and highly annoyed (score 8–10; 31.5). This categorization was based on a report of the Dutch National Institute for Public Health that used the same data [66]. The same pattern of higher average psychological distress was found for those who were highly annoyed by odour due to agriculture. However, for odour annoyance relating to other sources, those who were quite annoyed reported the highest levels of psychological distress (though the n of 13 individuals with high levels of this type of odour annoyance is small). In addition, before turning to our results from the regression models, we presented the correlations between noise and odour annoyance and psychological distress. The correlation between noise annoyance and psychological distress was 0.17, for odour due to agriculture it was 0.08 and for odour due to other sources it was 0.13.

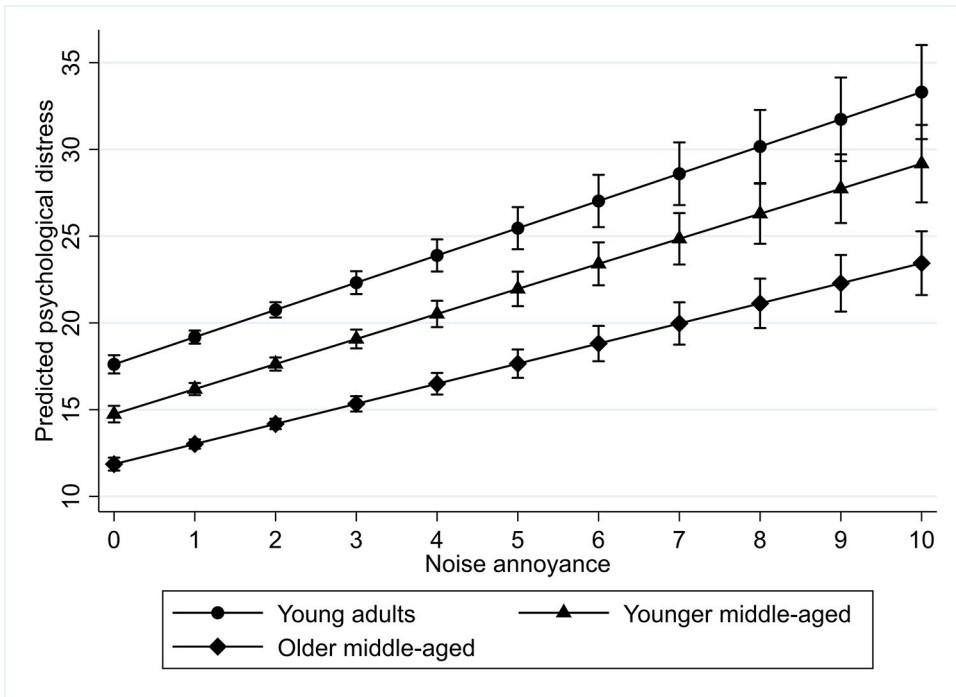

**Fig 1. Interaction plot for predicted psychological distress among different age groups by level of noise annoyance.**

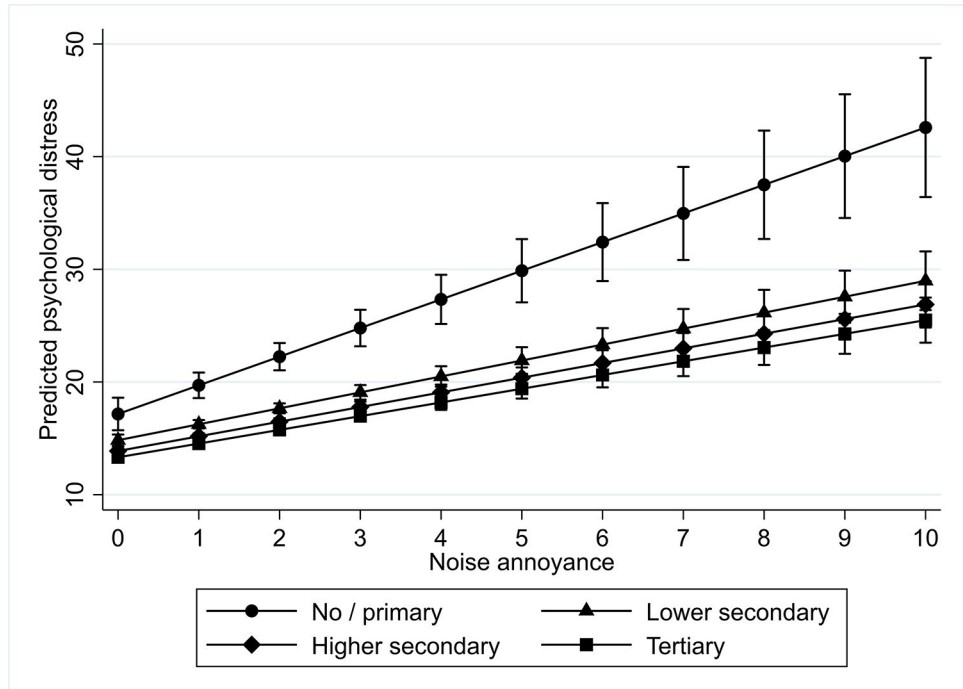

**Fig 2. Interaction plot for predicted psychological distress among different educational levels by level of noise annoyance.**

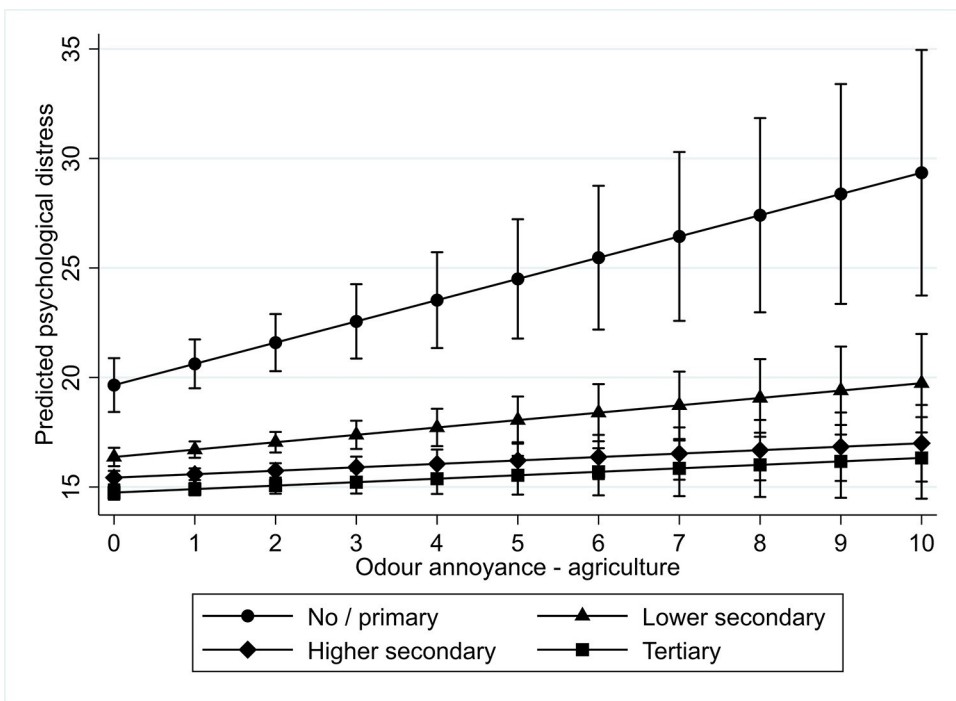

**Fig 3. Interaction plot for predicted psychological distress among different educational levels by level of odour annoyance by agriculture.**

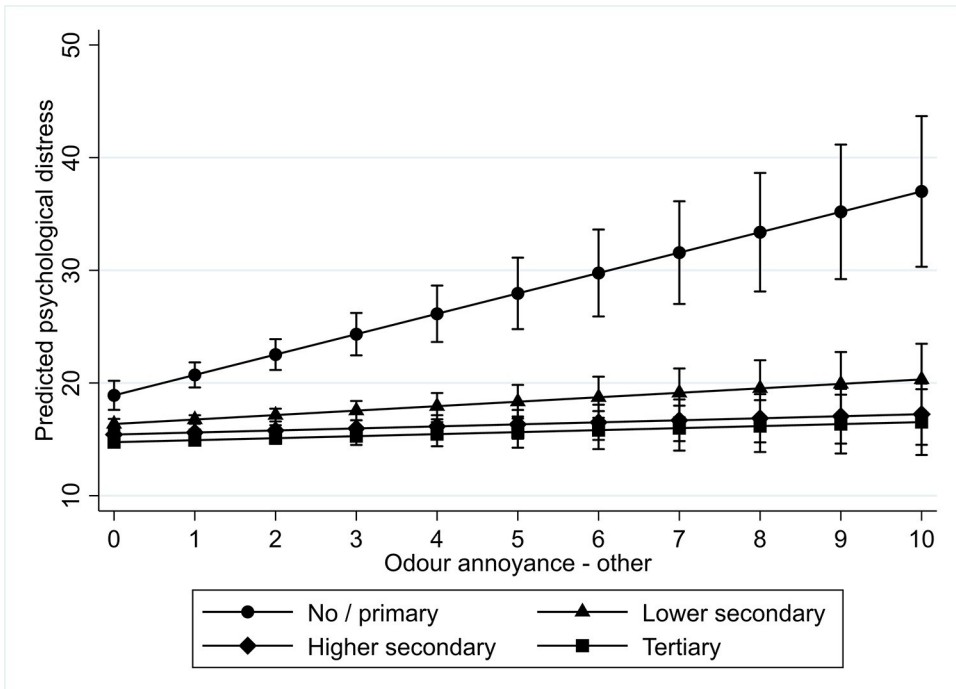

**Fig 4. Interaction plot for predicted psychological distress among different educational levels by level of noise annoyance by other sources.**

Turning to the results from our ordinary least squares regression, in Model 1 from Table 2, we found a positive and significant effect of noise annoyance on psychological distress. Since the psychological distress scale ranges from 0 to 100, the effects can be interpreted as percentage point changes. To make our findings more intuitive, we compared the psychological distress scores of individuals that were not annoyed at all by noise (scoring 0) to those that were most annoyed (scoring 10), and we found a difference in the predicted psychological distress score of 13.3 percentage points. In Model 2, both types of odour annoyance showed a positive and significant effect on the level of psychological distress. Individuals that were most annoyed by odour from agriculture reported an increase of 4.1 percentage points in psychological distress compared to those who were least annoyed. For odour annoyance from other sources the difference was larger: those who were most annoyed reported an increase of 13.6 percentage points in psychological distress compared to those who were least annoyed. In Model 3, all three sources of annoyance showed a positive effect on psychological distress, independent of each other, supporting the idea that they each have a unique contribution in explaining psychological distress among our adult population.

After testing the direct effects of noise and odour annoyance, we added different control variables to test the robustness of the results, as described in the Analytical strategy section. When comparing model 1 from Table 3 to model 3 in Table 2, we concluded that the effects of noise and odour annoyance on psychological distress are robust after adding parenthood. After adding self-rated health to the model, the effects of noise and odour annoyance on psychological distress became smaller (except for odour annoyance due to agriculture, where the coefficient remained stable). However, all three effects remained significant. Finally, in Table 3 model 3 we added both parenthood and self-rated health and in this model we again found significant positive effects for noise and odour annoyance on psychological distress.

Overall, these results supported hypothesis 1a and 1b, which stated that higher levels of noise and odour annoyance are associated with higher levels of psychological distress. When

comparing their relative contribution to the psychological distress of individuals, we found that noise annoyance had the largest standardized effect (beta = 0.11 compared to 0.02 for both sources of odour annoyance). This was in line with hypothesis 2, which stated that noise annoyance would have a stronger relative contribution to psychological distress than odour annoyance.

### Different relationships across sociodemographic groups

For noise annoyance, we found two significant interactions which are presented in Tables 4 and 5. We found a negative interaction of noise annoyance for older middle-aged adults (and young middle-aged, but this result was not significant) (see Table 4). This implies that, contrary to what we expected, among younger adults (compared to older middle-aged) noise annoyance was more strongly related to psychological distress. For younger adults the increase in psychological distress between those who were not annoyed and those who were most annoyed was 15.7 percentage points, whereas this was 14.4 and 11.6 percentage points respectively for the younger middle-aged and older middle-aged. In Fig 1 we present the predicted level of psychological distress for the different age groups at different levels of noise annoyance. Overall, these findings go against our hypothesis 4a, which stated that the relationship between noise annoyance and psychological distress would be stronger among older adults compared to their younger counterparts.

We found significant positive interaction effects for noise annoyance across different educational levels (as can be found in Table 5). The interactions showed differences in the psychological distress of those with no or primary education and those with tertiary education at similar levels of noise annoyance, but not between those with no or primary education and those with secondary education. This means that the psychological distress of those with no or primary education was higher when compared to those with tertiary education at similar levels of noise annoyance. This is also depicted graphically in Fig 2.

We found a similar result for odour annoyance. For both odour annoyance due to agriculture and to other sources, we found that those with no or primary education experience more psychological distress as a result of it compared to those with tertiary education. Figs 3 and 4 graphically present these results. All in all, we found support for hypothesis 5b, which stated that the psychological distress of less educated people would be stronger affected by odour annoyance than of better educated.

Finally, we found non-significant interactions. For noise annoyance, the psychological distress of women was not more strongly affected by noise annoyance compared to men's (hypothesis 3a) and for both types of odour annoyance we did not find significant interactions across gender or age groups. Hypotheses 3b and 4b did not receive thus support from our analyses.

### Additional analysis on the sample including older adults

We re-estimated the model including the interaction with age on the sample that included older adults (65+) (Tables A2-A5 in the S1 Appendix). Like in the previous analyses, we found a significant negative interaction between noise annoyance for young adults when compared to older middle-aged and this interaction effect was also present for older adults (results also presented in Fig 4 in the S1 Appendix).

### Sensitivity analyses

We re-estimated our results using logistic regression and a dichotomous dependent variable that differentiated between respondents at risk and not at risk of clinical depression and

anxiety disorder. We found that higher levels of noise annoyance and odour annoyance due to other sources were related to a higher chance for clinical depression and anxiety disorder. We did not find a significant effect for odour annoyance due to agriculture after controlling for the other two sources of annoyance (but there was a significant positive effect in the model that did not include noise annoyance). However, different from the main analyses, the interaction terms with age groups or educational levels were not significant.

## Conclusion and discussion

In line with previous studies linking noise and odour annoyance and mental health, we found that individuals who experienced higher levels of noise and odour annoyance (due to agriculture or other sources) in their home environment reported higher levels of psychological distress, even after controlling for regional variation and self-reported risks for exposure to noise and odour. When looking into the relative impact of noise and odour annoyance on psychological distress, and in line with a previous study [23], we found that the standardized effect of noise annoyance was the largest. Based on these findings, we encourage researchers and policy makers to pay closer attention to subjective perceptions of the environmental conditions (especially for noise annoyance), as these form a relevant indicator of the burden on health [45]. As such, the subjective evaluations of noise and odour provide a relevant starting point for public health strategies [25], e.g., problem areas within neighbourhoods can be identified or results of policy measures targeted at improving living conditions can be easily evaluated without the need of complex technology or prolonged data collection.

Based on the scarce literature on differential vulnerability to noise and odour annoyance among different sociodemographic groups [45, 60], we expected to find that these environmental conditions would be more harmful for women, individuals with lower levels of education and for older individuals. Lacking an established theoretical framework for such differences, we built our argumentation borrowing from theories and findings that regard differences in coping resources and daily living patterns in the population. Our reasoning was that these elements are important explanations of why noise and odour annoyance would impact psychological distress and their systematic unequal distribution among specific groups would translate into stronger or weaker effects.

Not all our results were in line with this reasoning. For example, we found that among older middle-aged adults noise annoyance was *less* strongly associated with psychological distress compared to their younger counterparts. Although seemingly counter intuitive, this finding is in line with Pai and Kim [51] who found that for older adults the relationship between neighbourhood perception of physical disorder (e.g., litter or air quality) and psychological distress is less strong compared to their younger counterparts. Their explanation was that older adults have more life experience and a greater psychological strength, which enables them to deal better with a disorderly environment. Moreover, because older adults more often experience hearing loss, they might be less affected, especially by noise annoyance [67].

Contrary to the study by Dratva, Zemp [45], we did not find gender differences in the effects of noise and odour annoyance on psychological distress. We argued that women have lower levels of coping resources such as personal control or financial resources compared to men, but this might not hold in a relatively gender equal society such as the Netherlands (ranked 4 out of 162 countries based on the 2019 Gender Inequality Index [68]). If so, when confronted with unpleasant environmental situations that result in noise and odour annoyance, Dutch women could feel that they can act in a way that would address the problem just as much as Dutch men. Moreover, we argued that women spend more time at home and this would translate in a stronger effect of their environment on their psychological distress. This

argument might not hold because the Netherlands has experienced an increase in the labour market participation of women over the years [69]. Finally, and in line with our expectations, we found that educational level was an important factor, i.e., the psychological distress of those with no or primary education was more strongly affected by noise annoyance and odour annoyance compared to those with a tertiary education.

These results present a scattered image of how noise and odour annoyance differently impact psychological distress. Due to data limitations, we were unable to examine the underlying mechanisms that we proposed and in the light of the mixed findings we encourage future research along these lines. In addition, future studies could take other elements of the neighbourhood into account next to noise and odour annoyance, for instance the presence of green space or neighbourhood facilities. This would provide a more in-depth picture into how different aspects of the living environment together influence the mental health of individuals [2]. Finally, as stated in the introduction, there are many variables that influence both noise and odour annoyance on the one hand and psychological distress on the other hand. We cannot rule out that underlying personality traits (like sensitivity to environmental factors) influence both noise annoyance and noise sensitivity [15]. Future studies should investigate this relationship. In addition, a longitudinal design could provide more clarity on the causal relationship between noise and odour annoyance and psychological distress.

To sum up, we provided evidence that both subjective perceptions of noise and odour annoyance are important for psychological distress, but when comparing their relative influence, noise annoyance had a stronger relative contribution to psychological distress. Evidence for stronger effects of noise and odour annoyance on the psychological distress across sociodemographic groups was mixed. In general, our results suggest that classic dimensions of social inequality along age and education lines are reinforced when focusing on the relationship between noise and odour annoyance.

## Supporting information

**S1 Appendix.**
(DOCX)

## Acknowledgments

We would like to thank Theo Kuunders, Leonard Vanbrabant and Joyce de Goede for their input in developing the research topic and their feedback on the manuscript.

## Author Contributions

**Conceptualization:** Eline Berkers, Ioana Pop, Mariëlle Cloïn, Antje Eugster.

**Formal analysis:** Eline Berkers.

**Project administration:** Mariëlle Cloïn.

**Supervision:** Ioana Pop, Mariëlle Cloïn, Hans van Oers.

**Writing – original draft:** Eline Berkers, Ioana Pop.

**Writing – review & editing:** Eline Berkers, Ioana Pop, Mariëlle Cloïn, Antje Eugster, Hans van Oers.

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
