## [Decision Letter · Decision Letter 0]

17 May 2021

PONE-D-21-06093

The relative effects of self-reported noise and odour annoyance on psychological distress: different effects across sociodemographic groups?

PLOS ONE

Dear Dr. Berkers,

Thank you for submitting your manuscript to PLOS ONE. After careful consideration, we feel that it has merit but does not fully meet PLOS ONE’s publication criteria as it currently stands. Therefore, we invite you to submit a revised version of the manuscript that addresses the points raised during the review process.

A **rebuttal letter** that responds to **EACH** point raised by the academic editor and reviewer(s). You should upload this letter as a separate file labeled 'Response to Reviewers'.A **marked-up copy** of your manuscript that highlights changes made to the original version. You should upload this as a separate file labeled 'Revised Manuscript with Track Changes'.An **unmarked version** of your revised paper without tracked changes. You should upload this as a separate file labeled 'Manuscript'.

We look forward to receiving your revised manuscript.

Kind regards,

Brecht Devleesschauwer

Academic Editor

PLOS ONE

Additional Editor Comments (if provided):

Both reviewers evaluated the manuscript in great detail, and provided valuable suggestions to improve the presentation and relevance of the manuscript. The reviewers also identified several major shortcomings of the paper, which should be addressed where possible, and otherwise carefully discussed in the limitations section of the manuscript.

In your revision note, please include EACH of the reviewer comments, provide your reply, and when relevant, include the modified/new text (or motivate why you decided not to modify the text). Note that failure to do so will result in a rejection of the manuscript.

Journal Requirements:

2) We note that you have indicated that data from this study are available upon request. PLOS only allows data to be available upon request if there are legal or ethical restrictions on sharing data publicly. For more information on unacceptable data access restrictions, please see http://journals.plos.org/plosone/s/data-availability#loc-unacceptable-data-access-restrictions.

3) Please ensure that you refer to Figure 5 in your text as, if accepted, production will need this reference to link the reader to the figure.

4) Please include a caption for Figure 5.

Reviewers' comments:

Reviewer's Responses to Questions

**Comments to the Author**

1. Is the manuscript technically sound, and do the data support the conclusions?

Reviewer #1: Yes

Reviewer #2: No

2. Has the statistical analysis been performed appropriately and rigorously? 

Reviewer #1: No

Reviewer #2: No

3. Have the authors made all data underlying the findings in their manuscript fully available?

Reviewer #1: No

Reviewer #2: No

4. Is the manuscript presented in an intelligible fashion and written in standard English?

Reviewer #1: Yes

Reviewer #2: Yes

5. Review Comments to the Author

Reviewer #1: The main added value of this research is to simultaneously model the association between noise and odor annoyance and psychological distress, and to assess the potential interaction effect with socio-demographic variables.

The claim is however not properly placed in the context of the previous literature. The background section lack of important references related to the burden of environmental noise and the mechanisms by which noise affect mental health.

Data and analysis support the claim. However, statistical analyses could be slightly improved. The statistical methods section should contain more details and the result section could be written in a clearer manner with regard to the results displayed in the table.

In general, the manuscript could be more concise and English style could be improved.

Abstract

*A repetition could be deleted in the abstract: “with noise annoyance having a relatively stronger effect than odor annoyance”)

* Statistical methods used could be mentioned in the abstract

Introduction

* Introduction should include information on the burden of psychological distress, depression and anxiety. Why is it an important public health problem?

* Also, references on the burden of environmental noise is lacking.

Clark, C., Paunovic, K., 2018. WHO Environmental Noise Guidelines for the European Region:

A Systematic Review on Environmental Noise and Quality of Life, Wellbeing and Mental

Health. International Journal of Environmental Research and Public Health 15, 2400.

World Health Organization . Burden of Disease from Environmental Noise. Quantification of Healthy Life Years Lost in Europe. World Health Organization; Copenhagen, Denmark: 2011

*Line 2, more recent references and systematic reviews on the association between urban environment and mental health could be added, such as:

Gruebner O, Rapp MA, Adli M, Kluge U, Galea S, Heinz A. Cities and mental health. Dtsch Arztebl Int. 2017;114(8):121–7.

Rautio N, Filatova S, Lehtiniemi H, Miettunen J. Living environment and its relationship to depressive mood: a systematic review. Int J Soc Psychiatry. 2018;64(1):92–103.

*The introduction lack of information on the non-auditory effect (and biological mechanisms) of noise on psychological distress. Two pathways are relevant for the development of adverse health effects of noise: the ‘direct’ and the ‘indirect’ effect. The ‘direct’ pathway is determined by the effect of noise of the central nervous system. The ‘indirect’ pathway refers to the cognitive perception of the sound and emotional responses such as annoyance. Both reaction chains may induce physiological stress.

“Environmental Exposures and Depression: Biological Mechanisms and Epidemiological Evidence | Annual Review of Public Health”

*Line 6: “In this study, psychological distress is defined as…”.

This is actually part of the method section.

*How is noise annoyance related to objective noise? It would be useful to introduce in this section the concept of noise sensitivity, which is known to modulate the association between noise and noise annoyance. In a review, van Kamp and Davies concluded that individuals with mental disorders constitute a risk group for noise sensitivity.

Schreckenberg, D., Griefahn, B., Meis, M., 2010. The associations between noise sensitivity, reported physical and mental health, perceived environmental quality, and noise annoyance.

Noise Health 12, 7–16. https://doi.org/10.4103/1463-1741.59995

Miedema H., Vos H. Noise sensitivity and reactions to noise and other environmental conditions. JASA. 2003;113:1492–1504.

Data and method

*Could you add a reference for the tool used to measure the level of psychological distress? (The Kessler 10 scale)

* The choice of the control variables should be better explained. Confounding variables are variables that affect both the exposure and the outcome.

Regarding the variable “Parenthood”, you explain how it can affect the level of annoyance but it is not clear how parenthood can affect psychological distress. For “Self-rated health”, you do not explain the reason why you decide to adjust for this variable? This variable is probably strongly associated with the outcome psychological distress. By adjusting for this variable, you probably lose a part of the effect. Give more details on how this question is asked? Does it include physical and mental health status?

*Noise annoyance. It is not clear for the reader how you treat the variable in your statistical models. As a dummy variable, with 10 dummies? With level 0 as reference ? Or on the continuous scale?

*Regarding the analytical strategy, I would perform different models with an increasing level of adjustment for the control variables. Do we obtain the same results without adjusting for parenthood and self-rated health?

*Did you take into account the sample design in the analysis?

Results

*Table 1

Descriptive statistics should part of the result section.

For noise and odor annoyance, it would be interesting to mention the number of cases by level ( if you treat the variable as a categorical variable in your analysis)

You mention the mean of the variable in the table. I guess then you included this variable on the continuous scale in the model? If it is the case, then the interpretation in the result section is confusing. Instead, the interpretation should be “ for one unit increase on the annoyance scale, the coefficient is…”

*Was Psychological distress normally distributed? Did you try to transform the outcome variable to approach normality? If the distribution of the variable (psychological distress or annoyance) is not normal, the median is more informative.

*In table 1

Take care of the terminology. Is the variable “Physical health” in the table 1 the “Self-rated health”?

* How was noise and odor annoyance include in the model? Continuous scale or dummy variable?

*Some interactions were significant. It would therefore make sense to present the models stratified by age and educational level. Is there a reason why you did not present the stratified models?

*You mention the results of bivariate analysis but this is not mentioned in the method section?

Also, in those results, annoyance seems to be categorized in “somewhat annoyed” and “highly annoyed”. Please, inform the reader about all the data transformation in the method section and the reason why you decide to categorize it in this way (cut off at 5?). If you decide to use the categorical variable for annoyance, then include the number of cases by category in the descriptive table.

* “When comparing the psychological distress scores of individuals that were not annoyed at all compare to that were most annoyed, we found a difference in predicted psychological distress of 13.8 points …..For odor annoyance from other sources the difference was larger: those who were most annoyed reported an increase of 11 percentage points in psychological distress ”

I do not find those results in the table 2. Are they visible in a figure?

Discussion

*All data included in the analysis are subjective variables and thus might be influenced by confounding variables. It is therefore plausible that there is a general vulnerability that renders people susceptible to report higher noise annoyance and mental health impairment . This should be mentioned in the discussion part.

Reviewer #2: GENERAL COMMENTS:

The study under review analyzed self-report data from a health monitoring survey in the Netherlands and comes to the conclusion that noise and odor annoyance are important indicators of psychological distress, with noise annoyance having a relatively stronger effect than odor annoyance. The "effects" (if one wants to call them so) found are small in my view (probably got significant only because the sample is relatively large). For example, psychological distress on a scale between 0 and 100 increases by just 3.8% between the least and the most annoyed from agricultural odor.

Firstly, I have some doubts whether the study conducted here really investigated a question relevant enough, that would justify a publication in PLOS ONE. Secondly, the study has some methodological shortcomings that I regard problematic.

For a start, the very research rationale does not become clear to me. For example, there is a wealth of studies into effects of environmental exposures on health outcomes, including annoyance, mental health and psychological distress. It don't see how the study under review really fits in here and if the very research question was addressed in previous work. Also, the concept of "annoyance" already expresses an emotional and/or behavioral reaction to some stressor and is therefore close to the concept of "psychological distress". I then does not seem to make much sense to report about a correlation between two constructs that are already relatively close to each other. Much more relevant to investigate would be a question like "what makes the people annoyed?".

Noise and odor annoyance are in a way treated as causal factors responsible for heightened levels of psychological distress, this becomes apparent e.g. in formulations like "more strongly affected", which I don't find very convincing. More convincing (and more useful for practical applications) would be to elucidate the relationship between noise exposure and noise annoyance (as a direct effect or mediator) and then its relationship with psychological distress.

What can be observed in such survey data is just correlations, which may allow causal inferences only when the sample is deliberately stratified to increase the contrast between not at all annoyed and extremely annoyed, and some sort of causality criteria applied. None of this was the case here. It is well possible that noise annoyance and psychological distress are both signs of some personality trait (e.g. general sensitivity to environmental factors, in particular noise sensitivity) that is responsible for both effects. Under that assumption, it is no surprise that both are related to each other. However the authors make no effort to discuss such mechanisms as a potential alternative explanation for their findings.

To cut a long story short, the survey sample lacks the collection of conceptionally important additional variables, namely environmental exposure data on one hand, in the present case, noise exposure from roads, railways, and aircraft (e.g. from so called noise maps), and noise sensitivity data on the other (e.g. measured with the Weinstein scale). This would have opened a more interesting avenue to data analysis, e.g. the possibility to answer the question if noise annoyance is a mediator or moderator variable in the link between noise exposure and psychological distress.

I therefore suggest to reject the manuscript.

In the following, I list further comments page by page.

COMMENTS BY SECTION:

Page 2, Abstract: "distress and." doesn't make sense.

Page 3: Some of the sentences in the text are not very precisely worded in my view. E.g. Page 3, 1st paragraph: a "stressor" cannot be a "reaction" at the same time. One does not "model noise annoyance and odor annoyance" at the same time, but psychological distress (because psychological distress is the dependent variable here).

Page 5: The disquisition about the theoretical background is a bit too long for my taste.

Page 6: I don't understand this sentence: "Secondly, regarding the relative impact of noise and odor annoyance on psychological distress, we expect that for odor annoyance there is more variation in the levels of annoyance in comparison to noise (5)."

Page 7, 2nd paragraph: Again, too many textbook-style phrases. I would considerably shorten this part.

Page 9, the following doesn't make sense to me: "Additionally, to test the relationship between noise and odor annoyance across more age groups, we use a larger sample which includes also older adults age 65+ (n = 34838, of which 25326 adults and 9512 older adults). We did not use this sample in our main analysis because these respondents only answered a subset of the items in the noise annoyance scale (three out of nine) and all respondents originated from the same subregion (South-East Brabant)." Why do authors mention this if they don't use that sample?

Page 10, first paragraph: Authors should give some more information about the sampling procedures that were applied... (and maybe also some info about the background and the goals of the health monitoring project as a whole).

Page 10, first paragraph: It seems to me important to mention the percentage of deleted records and to discuss any missing data analysis options that were considered. Were the data missing at random? Why was imputation not considered? Btw. the correct sentence here should read. "After deleting the cases with missing values on the main variables, the final number of (analyzable) cases was 25236." -- you cannot 'delete' missing values (as your version of the sentence does), but only cases with missing values. (There are quite a few similar language glitches in the text)

Page 11, Noise Annoyance: While there is no "objective" metric for measuring odors, it should have been possible to assign noise exposure values for road, rail, and aircraft noise to each of the respondents, e.g. with the use of national noise maps (that surely must exist for the Netherlands). Authors should explain why they did not include noise exposure data.

Page 11, Odor Annoyance: Authors should mention how the odor question was exactly posed, and how many respondents felt such annoyances at their home. I can imagine that the majority of the sample did not sense any disturbing odors in their living environment, i.e. the place they spend most of their time.

Page 13, first line: something is missing here.

Page 13: dummies = either 'dummy variable' or 'indicator variable'

Page 14 - Analytical strategy: Earlier in the manuscript, authors mention that they have a few "hypotheses". If hypotheses are involved, then the alpha level adopted for the statistical analyses should also be given.

Page 15: Reformulate "Turning to the results from our ordinary least squares regression, in Model 1 from Table 2, we present a positive and significant effect of noise annoyance on psychological distress." Rather write "we found a positive and significant effect...".

Page 16, Model 3: The analyses presented here all call for a thorough analysis of potential multicollinearity between the main predictors, especially because the authors mention that they are "independent". VIF's (variance inflation factors) should be estimated and included in the results. A large VIF on an independent variable indicates a highly collinear relationship to the other variables that should be considered or adjusted for in some way (e.g. by excluding 'double counting' variables from the model)

Table 2: Footnote for "Beta" ("Beta" is probably Psychology jargon... and might not be understood by people from other disciplines..), Also explain "R2"

Table 2 and 3: Indicate p-values in a separate column, there are no reasons why p-values are not given in full.

6. PLOS authors have the option to publish the peer review history of their article (what does this mean?). If published, this will include your full peer review and any attached files.

Reviewer #1: No

Reviewer #2: No

---

## [Author Response · Author response to Decision Letter 0]

20 Jul 2021

We have added a word document with the replies to all the comments previously (in a file called revision letter Plos one final). But all answers from that file are also pasted below:

We thank the editor for the further instructions on how to prepare the manuscript according to the journal’s requirements. The names of the separate files which included the figures were not according to PLOS ONE’s standards and are changed from ‘Figure 1’ into ‘Fig 1’. In addition, we changed the figure and table captions in bold font and carefully checked the headings of all paragraphs. 

Below we list fragments from the text. The revised text is highlighted in yellow.

Subheader: Additional analysis on the sample including older adults (page 24)

Table 1. Descriptive statistics of all variables (n = 25236) (page 15) 

Fig 1. Interaction plot for predicted psychological distress among different educational levels by level of noise annoyance (page 22)

In the revised title page, we used superscripts to indicate the institution instead of subscripts and indicated the sets of contributors who did an equal amount of work. See detailed below:

Eline Berkers1,2¶, Ioana Pop1¶, Mariëlle Cloïn2&, Antje Eugster2,3& and Hans van Oers2,4&

2) We note that you have indicated that data from this study are available upon request. PLOS only allows data to be available upon request if there are legal or ethical restrictions on sharing data publicly. For more information on unacceptable data access restrictions, please see http://journals.plos.org/plosone/s/data-availability#loc-unacceptable-data-access-restrictions.

The data cannot be shared due to legal restrictions. The data are owned by a third-party organization, i.e. the three local regional health service organizations that serve as partner organizations in this research.

Data are available from the website of the GGD GHOR if researchers meet the criteria. Data requests can be done at: https://monitorgezondheid.nl/contact or by email to monitorgezondheid@ggdghor.nl. 

3) Please ensure that you refer to Figure 5 in your text as, if accepted, production will need this reference to link the reader to the figure.

We have entered a reference to Figure 5 (now Figure 8) in the text on page 24. 

4) Please include a caption for Figure 5.

We have now included a caption for Figure 8 on page 35:

Fig 8. Interaction plot for predicted psychological distress among different age groups by level of subjective noise annoyance

Reviewer #1: 

The main added value of this research is to simultaneously model the association between noise and odor annoyance and psychological distress, and to assess the potential interaction effect with socio-demographic variables.

The claim is however not properly placed in the context of the previous literature. The background section lack of important references related to the burden of environmental noise and the mechanisms by which noise affect mental health.

We thank the reviewer for the suggestion and for the useful literature suggestions. We have included the references in the revised version of the manuscript and we now add more context from the literature on why noise and odour are important issues and what the mechanisms are that link noise/odour and mental health. In the first version of the paper, this was mostly lacking, as we wanted to make a clear focus on perceptions of noise and odour annoyance (as opposed to objective exposure to noise or odour). We recognize that this focus came across as too narrow. With the added literature, the revised manuscript makes clearer the position of our work within the broader literature. 

Below, we present the first paragraph of the revised manuscript that is revised with this suggestion in mind (revisions highlighted in yellow):

“Over the years, the mental health disadvantages of living in an unruly environment filled with environmental stressors have become clear (Gruebner et al., 2017; Honold, Beyer, Lakes, & van der Meer, 2012; Rautio, Filatova, Lehtiniemi, & Miettunen, 2018; Ross & Mirowsky, 2009; Schafer & Upenieks, 2015). Two of these stressors that negatively affect mental health are noise and odour. The WHO Regional Office for Europe (2011) estimates that every year more than 1 million healthy years are lost due to traffic noise only. (page 3)

In addition, we have added more information on the mechanisms that link noise and odour to mental health: 

“Noise and odour can be quantified by using decibels and odour standards (for example, ammonia thresholds) and are thought to affect individual’s mental health in two different ways: a direct pathway, i.e., a negative effect on the nervous system through physical arousal and an activation of stress hormones, and an indirect pathway through triggering an negative emotional responses such as anger or stress (Clark & Paunovic, 2018; Schiffman & Williams, 2005; van den Bosch & Meyer-Lindenberg, 2019).” (page 3)

Data and analysis support the claim. However, statistical analyses could be slightly improved. The statistical methods section should contain more details and the result section could be written in a clearer manner with regard to the results displayed in the table.

In general, the manuscript could be more concise and English style could be improved.

We have revised the data and results section (as will be detailed in the rest of this letter) and carefully reviewed the manuscript for brevity and clarity. 

Abstract

*A repetition could be deleted in the abstract: “with noise annoyance having a relatively stronger effect than odor annoyance”)

We have deleted the sentence “We conclude that noise and odour annoyance are important indicators of psychological distress, with noise annoyance having a relatively stronger effect than odour annoyance “ as it was indeed a repetition. 

* Statistical methods used could be mentioned in the abstract

We have now included this in the abstract: 

Using data from the Health Monitor (n = 25236) in Noord-Brabant, we found using Ordinary Least Squares Regression that individuals that reported higher levels of noise and odour annoyance reported higher levels of psychological distress. 

Introduction

* Introduction should include information on the burden of psychological distress, depression and anxiety. Why is it an important public health problem?

We have added references that identify why indeed mental health issues form an important public health problem on page 3: 

If we add to this the fact that mental disorders, and especially mood disorders, pose a very high burden on the population of Europe and are expected to become even more prominent for population health in the future (OECD & European Union, 2020; WHO Regional Office for Europe, 2015), it is clear that a better understanding of the relationship between odour and noise and mental health is needed in order to mitigate these noxious pathways.

* Also, references on the burden of environmental noise is lacking.

Clark, C., Paunovic, K., 2018. WHO Environmental Noise Guidelines for the European Region:

A Systematic Review on Environmental Noise and Quality of Life, Wellbeing and Mental

Health. International Journal of Environmental Research and Public Health 15, 2400.

World Health Organization . Burden of Disease from Environmental Noise. Quantification of Healthy Life Years Lost in Europe. World Health Organization; Copenhagen, Denmark: 2011

We thank the reviewer for these valuable suggestions and accordingly have rewritten the first part of the introduction. It now includes the suggested references and similar references for odour to indicate the burden of these stressors.

The WHO Regional Office for Europe (2011) for instance, estimates that every year more than 1 million healthy years are lost due to traffic noise only. In addition, odour pollution is associated with many health complaints and decreased quality of life (D-NOSES Consortium, 2019).

And on page 3: 

“Noise and odour can be quantified by using decibels and odour standards (for example, ammonia thresholds) and are thought to affect individual’s mental health in two different ways: a direct pathway, i.e., a negative effect on the nervous system through physical arousal and an activation of stress hormones, and an indirect pathway through triggering an negative emotional responses such as anger or stress (Clark & Paunovic, 2018; Schiffman & Williams, 2005; van den Bosch & Meyer-Lindenberg, 2019).”

*Line 2, more recent references and systematic reviews on the association between urban environment and mental health could be added, such as:

Gruebner O, Rapp MA, Adli M, Kluge U, Galea S, Heinz A. Cities and mental health. Dtsch Arztebl Int. 2017;114(8):121–7.

Rautio N, Filatova S, Lehtiniemi H, Miettunen J. Living environment and its relationship to depressive mood: a systematic review. Int J Soc Psychiatry. 2018;64(1):92–103.

We thank the reviewer for suggesting these useful references, which were added to paragraph 1 of our introduction. 

“Over the years, the mental health disadvantages of living in an unruly environment filled with environmental stressors have become clear (Gruebner et al., 2017; Honold et al., 2012; Rautio et al., 2018; Ross & Mirowsky, 2009; Schafer & Upenieks, 2015).” (page 3)

*The introduction lack of information on the non-auditory effect (and biological mechanisms) of noise on psychological distress. Two pathways are relevant for the development of adverse health effects of noise: the ‘direct’ and the ‘indirect’ effect. The ‘direct’ pathway is determined by the effect of noise of the central nervous system. The ‘indirect’ pathway refers to the cognitive perception of the sound and emotional responses such as annoyance. Both reaction chains may induce physiological stress.

“Environmental Exposures and Depression: Biological Mechanisms and Epidemiological Evidence | Annual Review of Public Health”

As suggested, we have now included these explanatory pathways in the first paragraph of the introduction. 

“Noise and odour can be quantified by using decibels and odour standards (for example, ammonia thresholds) and are thought to affect individual’s mental health in two different ways: a direct pathway, i.e., a negative effect on the nervous system through physical arousal and an activation of stress hormones, and an indirect pathway through triggering an negative emotional responses such as anger or stress (Clark & Paunovic, 2018; Schiffman & Williams, 2005; van den Bosch & Meyer-Lindenberg, 2019).” (page 3)

*Line 6: “In this study, psychological distress is defined as…”.

This is actually part of the method section.

In the previous version of the manuscript, this sentence had the role of making clear that our mental health outcome is psychological distress, a more precise concept than the generic “mental illness”. In the revised version of the introduction, we introduce this focus in a different way (see excerpt below) and in the methods section we provide the conceptualization of the “psychological distress” concept.

Furthremore, subjective perceptions are the pathway between objective stressors and the resulting emotional states (i.e., psychological distress) compared to objective measures (Hammersen, Niemann, & Hoebel, 2016; Lazarus & Folkman, 1984). (page 4)

*How is noise annoyance related to objective noise? It would be useful to introduce in this section the concept of noise sensitivity, which is known to modulate the association between noise and noise annoyance. In a review, van Kamp and Davies concluded that individuals with mental disorders constitute a risk group for noise sensitivity.

Schreckenberg, D., Griefahn, B., Meis, M., 2010. The associations between noise sensitivity, reported physical and mental health, perceived environmental quality, and noise annoyance.

Noise Health 12, 7–16. https://doi.org/10.4103/1463-1741.59995

Miedema H., Vos H. Noise sensitivity and reactions to noise and other environmental conditions. JASA. 2003;113:1492–1504.

Thanks to the reviewers suggestions, we realize that how noise annoyance is related to objective noise was underemphasized in the manuscript. Therefore, we revised this in the second paragraph of the introduction. In general, research has shown that about 1/3 of the variance in noise annoyance is explained by exposure to noise (Birk et al., 2011). We have clarified this in a new paragraph on page 3: 

Noise and odour can be quantified by using decibels and odour standards (for example, ammonia thresholds) and are thought to affect individual’s mental health in two different ways: a direct pathway, i.e., a negative effect on the nervous system through physical arousal and an activation of stress hormones, and an indirect pathway through triggering an negative emotional responses such as anger or stress (Clark & Paunovic, 2018; Schiffman & Williams, 2005; van den Bosch & Meyer-Lindenberg, 2019). However, people are differently affected by objective levels of noise or odour: what is annoying to some may not be to others depending on personal, social or general characteristics, such as personality, sensitivity to noise or odour, relationship with the source, ability to ‘escape’ and history of disturbance due to noise or odour (Birk, Ivina, von Klot, Babisch, & Heinrich, 2011; Miedema & Vos, 2003; Schreckenberg, Griefahn, & Meis, 2010). Subsequently, the relationship between objective noise exposure and noise annoyance is not that clear-cut and only around ⅓ of the variance in noise annoyance is explained by exposure to noise (Birk et al, 2011). Similar sources are not available for odour but it is reasonable to assume a similar association. 

In addition, we have added noise sensitivity as a possible explanation in the conclusion on page 27. 

Finally, as stated in the introduction, there are many variables that influence both noise and odour annoyance on the one hand and psychological distress on the other hand. We cannot rule out that underlying personality traits (like sensitivity to environmental factors) influence both noise annoyance and noise sensitivity (Schreckenberg et al., 2010). Future studies should investigate this relationship in more detail. 

Data and method

*Could you add a reference for the tool used to measure the level of psychological distress? (The Kessler 10 scale)

We added two references on the Kessler 10-scale: 

The survey included the Kessler-10 scale, which is commonly used to measure nonspecific psychological distress (e.g., symptoms of depression and anxiety disorders) (Donker et al., 2010; Kessler & Mroczek, 1994).

* The choice of the control variables should be better explained. Confounding variables are variables that affect both the exposure and the outcome.

We explained which controls were included and better argue this in the manuscript. 

We have included the following revisions in the data and methods section on page 14: 

Next, parents' child-rearing duties and work-life balance take up time and energy and consequently leads to higher levels of psychological distress (Nomaguchi & Milkie, 2003; Simon & Caputo, 2018; Umberson et al., 2010).

In addition, we controlled for self-rated health, because on the one hand, worse health increases the vulnerability for environmental stressors (Schreckenberg et al., 2010) and on the other hand, is strongly related to mental health (Boardman et al., 2008; Hammersen et al., 2016; Jensen et al., 2018). 

Regarding the variable “Parenthood”, you explain how it can affect the level of annoyance but it is not clear how parenthood can affect psychological distress. 

From the literature, the argument for including parenthood related to psychological distress is that the mental coping resources of parents with (young) children are more limited: the constraints and demands that are typically placed on the time and energy of parents when raising (young) children tax parents’ coping resources. Because of this, parents more often report symptoms of ill mental health compared to their childless counterparts (Nomaguchi & Milkie, 2003; Simon & Caputo, 2019; Umberson, 2010).

For “Self-rated health”, you do not explain the reason why you decide to adjust for this variable? This variable is probably strongly associated with the outcome psychological distress. By adjusting for this variable, you probably lose a part of the effect. 

By including self-rated health, we follow the existing literature, see for instance: Hammersen et al., 2016; Jensen et al, 2018. Previous literature shows that noise and odour sensitivity (and subsequently, also the level of annoyance due to these environmental factors) is related to health status (Schreckenberg, Griefahn & Meis, 2010). Our data also supports this relationship - those who are in good health report on average the lowest levels of noise and odour annoyance. In addition, self-rated health is indicative of psychological distress as poor self-rated health is a chronic life stressor (Boardman et al, 2008). These arguments support the inclusion of SRH as a control variable.

Yet we do agree with the reviewer that including SRH could partly capture the effect of noise and odour annoyance on psychological distress, because SRH could also play the role of a mediator. Objective noise and odour circumstances could directly affect physical health through direct physiological effects or by disturbing sleep / rest. A low SRH could become a distress cause for individuals. 

As such, by including SRH in our models, we believe we provide a more conservative test of our hypotheses, which only increases our confidence in the robustness of our conclusions.

Give more details on how this question is asked? Does it include physical and mental health status?

The specific question concerning self-rated health is: ‘in general, how do you perceive your health?’. Whether this refers to mental or physical health is not specified in the question. Answer options ranged between very good, good, average, bad, and very bad. We merged good and very good and very bad and bad into one category, thereby having 3 categories: good, average and bad. 

We have clarified this in the data and methods section: 

Self-rated health was measured by the following question: how do you perceive your health in general? (Idler & Benjamin, 1997). Three answer options were: bad, moderate and good health (ref.).

*Noise annoyance. It is not clear for the reader how you treat the variable in your statistical models. As a dummy variable, with 10 dummies? With level 0 as reference? Or on the continuous scale?

We treat both noise and odour annoyance as continuous variables. We have revised the description of data for clarity. 

Like previous studies that took into account multiple sources of noise (Dzhambov et al., 2018; Jensen et al., 2018), a mean score was computed based on a minimum of eight out of the nine items (n = 352 individuals answered only eight items). For our analyses, we used the scale as a continuous variable ranging from 0 “no odour annoyance” to 10 “high odour annoyance”.

*Regarding the analytical strategy, I would perform different models with an increasing level of adjustment for the control variables. Do we obtain the same results without adjusting for parenthood and self-rated health?

Following this suggestion, we now present Table 2 and 3 with a sequence of nested models. In Table 2, we present model 1 which includes the sociodemographic and environmental control variables and noise annoyance. Next, model 2 includes the sociodemographic control variables and the two types of odour annoyance. And finally model 3 includes the sociodemographic control variables and noise and odour annoyance. In Table 3, we also added separately to the models parenthood in model 1, and self-rated health in model 2. Model 3 includes all control variables including parenthood and self-rated health. We base all interactions on model 3. 

This strategy has also been explained in the manuscript on page 16. 

When comparing the models where parenthood was and was not included (model 1 in Table 3 versus model 3 in Table 2), our results were robust and there was little change in the coefficients.

When comparing Model 1 and Model 2 in Table 3, we noted that including SRH explained away part of the relationship between noise and odour annoyance variables and psychological distress However, these effects were still significant and our conclusions did not change. As such, including SRH in the models as a control variable provides a stricter test of the hypotheses.

*Did you take into account the sample design in the analysis?

The sample we used is a random sample based on the Municipal Records database taken by Statistics Netherlands (the national statistical office of the Netherlands), and covers individuals that live independently aged between 19 and 64. 

In our OLS regression models, we did not use sample weights because we are not interested in determining accurate population estimates but we are interested in estimating relationships. For such a task, the question of using or not sample weights is not a straightforward one (see for example this explanation on the blog on the site of the World Bank, based on a paper by Solon, Haider & Wooldridge: https://blogs.worldbank.org/impactevaluations/tools-of-the-trade-when-to-use-those-sample-weights). We preferred to improve the robustness of our results by estimating: 1) additional analyses for the sample of older adults and 2) sensitivity analyses as presented in the manuscript. Furthermore, as discussed above, we controlled our models as rigorously as possible with the data at hand and examined whether different methodological specifications have a significant impact on our conclusions. We believe that such an approach is most transparent, when the task is to provide more certainty into the robustness of our results and conclusions. 

Results

*Table 1

Descriptive statistics should part of the result section.

We thank the reviewer for the suggestion. We decided to leave Table 1 in the Data and Methods section. In the revised version, we have moved it from the end of the Data and Methods section to page 14/15 (after the discussion of the conceptualization of the variables and before the Analytical strategy section). This table provides the quantitative information that goes together with the operationalization of the variables as described in the text and so, we find it most logical to have it inserted there. The table does not contain substantive results, but merely provides an overview of the available data and their structure. For this reason we do not see the added value of including the table in the result section.

We do indeed start the Results section with a brief description of bivariate analyses but these are summarized in the Appendix, as indicated in the text. 

For noise and odor annoyance, it would be interesting to mention the number of cases by level ( if you treat the variable as a categorical variable in your analysis).

In all our formal tests, we treated noise and odour annoyance as continuous variables. However, following the suggestion of the reviewer, we decided to add histograms in the Appendixes (Fig 5 to 7) in order to more transparently show the distribution of the variables measuring the odour and noise annoyance. 

You mention the mean of the variable in the table. I guess then you included this variable on the continuous scale in the model? If it is the case, then the interpretation in the result section is confusing. Instead, the interpretation should be “ for one unit increase on the annoyance scale, the coefficient is…”

We interpret this comment as a result of the confusion created by the previous version of the manuscript, when it was not clear that in our formal tests, the variables measuring odour and noise annoyance were used as continuous variables with a range from 0 to 10. We have clarified this in the text, as described in a previous answer.

Next, the dependent variable was rescaled from 0 to 100. Our reason to do so was that we can interpret the effects as percentage change, i.e., for one unit increase in the annoyance scale, we observed a x percent change on the psychological distress scale. We have added a sentence in the text to clarify our strategy: 

We computed a sum score, but to ease the interpretation of the results we rescaled the scale to range between 0 and 100, where a higher score represented higher levels of psychological distress. Subsequently, we can interpret the coefficients from our regression models as changes in percentage points. (page 11)

In addition, we mention that the way we present the results was meant to facilitate the estimation of the strengths of the effects, by comparing the differences in psychological distress between the individuals with minimum score on the annoyance scales with those with maximum score. Because these differences are expressed in percentage points, this comparison is more meaningful and intuitive than the coefficient itself and also avoids the repetition of the information that is already present in the table.

*Was Psychological distress normally distributed? Did you try to transform the outcome variable to approach normality? If the distribution of the variable (psychological distress or annoyance) is not normal, the median is more informative.

The variable measuring psychological distress was not normally distributed. We have tested a log transformation and estimated our models using it. In general, all our conclusions were robust. Because the coefficients are easier to be interpreted using the not-transformed dependent variable, we opted to use it as is. 

*In table 1

Take care of the terminology. Is the variable “Physical health” in the table 1 the “Self-rated health”?

The reviewer is correct that we used the wrong term here. We have changed ‘physical health’ to ‘self-rated health’. 

* How was noise and odor annoyance include in the model? Continuous scale or dummy variable?

We used the scales measuring noise and odour annoyance as continuous variables. We have clarified this in the text, as explained above.

*Some interactions were significant. It would therefore make sense to present the models stratified by age and educational level. Is there a reason why you did not present the stratified models?

Opting to present models stratified by age and education level would present the coefficients of the annoyance indicators by the categories of the above mentioned variables but would not inform the reader whether the differences between coefficients are statistically significant. As such, we preferred to present the interaction models together with the interaction plot, for a visual intuitive understanding of the significant differences found. 

*You mention the results of bivariate analysis but this is not mentioned in the method section?

We added this to the method section on page 16 (and added additional information about the categories to the results section, as shown in the next answer): 

For our bivariate analysis, we showed the psychological distress scores of those with different scores on the noise and odour annoyance scale to illustrate if there are differences between these groups. 

Also, in those results, annoyance seems to be categorized in “somewhat annoyed” and “highly annoyed”. Please, inform the reader about all the data transformation in the method section and the reason why you decide to categorize it in this way (cut off at 5?). If you decide to use the categorical variable for annoyance, then include the number of cases by category in the descriptive table.

This data transformation (using categories) was used only for this bivariate analysis with the aim of showing that individuals that are more annoyed as a result of noise or odour show higher levels of psychological distress. We have slightly changed the categories in order to clear up the confusion about the cut-off points. Our reasoning is explained below: 

Simple bivariate analyses (Table A1 in the appendix) showed that those who were least annoyed by noise had the lowest average psychological distress score (score 0-2; 14.9) as compared to those who were annoyed (score 3-7; 21.1) and highly annoyed (score 8-10; 31.5). These categories were also used in a report of the Dutch National Institute for Public Health that relied on the same data (Slob et al., 2019).

* “When comparing the psychological distress scores of individuals that were not annoyed at all compare to that were most annoyed, we found a difference in predicted psychological distress of 13.8 points …..For odor annoyance from other sources the difference was larger: those who were most annoyed reported an increase of 11 percentage points in psychological distress ”

I do not find those results in the table 2. Are they visible in a figure?

These results were predicted scores based on the regression estimates derived from Model 1 in Table 2. We have clarified this finding in the text on page 17: 

Turning to the results from our ordinary least squares regression, in Model 1 from Table 2, we found a positive and significant effect of noise annoyance on psychological distress. Since the psychological distress scale ranges from 0 to 100, the effects can be interpreted as percentage point changes. To make our findings more intuitive, we compared the psychological distress scores of individuals that were not annoyed at all by noise (scoring 0) to those that were most annoyed (scoring 10), and we found a difference in predicted psychological distress score of 13.3 percentage points. In Model 2, both types of odour annoyance showed a positive and significant effect on the level of psychological distress. Individuals that were most annoyed by odour from agriculture reported an increase of 4.1 percentage points in psychological distress compared to those who were least annoyed. For odour annoyance from other sources the difference was larger: those who were most annoyed reported an increase of 14.2 percentage points in psychological distress compared to those who were least annoyed.

Discussion

*All data included in the analysis are subjective variables and thus might be influenced by confounding variables. It is therefore plausible that there is a general vulnerability that renders people susceptible to report higher noise annoyance and mental health impairment . This should be mentioned in the discussion part.

This is indeed a relevant notion for the discussion. We have included a general sensitivity to environmental stressors to our results in the conclusion on page 27: 

Finally, as stated in the introduction, there are many variables that influence both noise and odour annoyance on the one hand and psychological distress on the other hand. We cannot rule out that underlying personality traits (like sensitivity to environmental factors) influence both noise annoyance and noise sensitivity (Van Kamp & Davies, 2013). Future studies should investigate this relationship in more detail.

Reviewer #2: GENERAL COMMENTS:

The study under review analyzed self-report data from a health monitoring survey in the Netherlands and comes to the conclusion that noise and odor annoyance are important indicators of psychological distress, with noise annoyance having a relatively stronger effect than odor annoyance. The "effects" (if one wants to call them so) found are small in my view (probably got significant only because the sample is relatively large). For example, psychological distress on a scale between 0 and 100 increases by just 3.8% between the least and the most annoyed from agricultural odor.

Our effects are indeed relatively small and this is to be expected for the following reasons. To start with, psychological distress is impacted by a myriad of factors (e.g., health status, critical life events, socio-economic standing, quality and quantity of social relationships, etc). Among these factors, environmental factors are relevant but not as important in terms of their impact as for instance losing a relative or becoming a parent. In addition, concerning environmental factors, odour and noise are most likely to play a modest role, especially within the context of the current situation in the Netherlands, where acceptable levels of noise and odour are regulated by law. Still even regardless of these regulations, noise and odour remain important environmental issues that could be addressed by policy makers to improve public health.

The fact that such factors are small does not mean that they are insignificant for people’s lives. Life is full of such low-impact factors (the daily hustles) that add up and use the coping resources available for people. Their added force combined can be substantial for individuals, especially because they tend to concentrate in certain areas and groups. With this in mind, we argue that investigating such low-impact factors, and especially potential systematic inequalities in these factors among social groups adds up to understanding the general inequalities in health and wellbeing. 

Firstly, I have some doubts whether the study conducted here really investigated a question relevant enough, that would justify a publication in PLOS ONE. Secondly, the study has some methodological shortcomings that I regard problematic.

With our previous answer, we hope to have touched on the problem addressed and relevance of our study. Below we address the observations presented by the reviewer.

For a start, the very research rationale does not become clear to me. For example, there is a wealth of studies into effects of environmental exposures on health outcomes, including annoyance, mental health and psychological distress. It don't see how the study under review really fits in here and if the very research question was addressed in previous work. 

Indeed, studies that investigate effects of environmental exposure on health outcomes have multiplied especially after the development of more sophisticated methods of measurement and analysis of such data. However, for practitioners as the policy makers at a municipality, such data is hard to understand, interpret and translate into policy. Furthermore, there is a difference between individuals in how they perceive such environmental influence (for instance because of differential sensitivity, as mentioned by reviewer #1). As such, looking at subjective measures of environmental influences such as noise and odour makes sense and is relevant for practitioners who routinely collect such information and want to understand whether this data can be useful in their work crafting policies directed at improving the wellbeing of citizens. 

Next, the question whether odour and noise annoyance similarly impacts different groups of people helps towards better understanding of the causes of health inequalities between populations. As mentioned above, such an endeavour is relevant because of the selection within specific neighborhoods which results in a clustering of risk factors for residents. Understanding if odour and noise annoyance also contributes to the general health inequalities due to this selection effect is, in our opinion, a valuable and relevant question both for science and society. 

Also, the concept of "annoyance" already expresses an emotional and/or behavioral reaction to some stressor and is therefore close to the concept of "psychological distress". It then does not seem to make much sense to report about a correlation between two constructs that are already relatively close to each other.

While we agree with the statement that both annoyance and psychological distress concepts reflect a subjective negative reaction to one’s environment / life situation, in our data the annoyance scales and psychological distress are not highly correlated. 

We have added the correlations to the manuscript on page 17:

In addition, before turning to our results from the regression models, we presented the correlations between noise and odour annoyance and psychological distress. The correlation between noise annoyance and psychological distress was 0.17, for odour due to agriculture it was 0.08 and for odour due to other sources it was 0.13.

Much more relevant to investigate would be a question like "what makes the people annoyed?".

If we understand the comment correctly, the reviewer’s suggestion is to focus better on understanding the (objective or subjective, environmental and personal) conditions that influence the subjective appraisal of noise and odour by individuals. Following the comments by reviewer#1, such a condition would be personal sensitivity. Of course, the objective levels of odour and noise would be another category of such precedents. 

We agree with the reviewer that such a question is relevant and we previously presented our arguments as to why our chosen research questions have practical and scientific relevance.

Noise and odor annoyance are in a way treated as causal factors responsible for heightened levels of psychological distress, this becomes apparent e.g. in formulations like "more strongly affected", which I don't find very convincing. More convincing (and more useful for practical applications) would be to elucidate the relationship between noise exposure and noise annoyance (as a direct effect or mediator) and then its relationship with psychological distress.

The relationships that the reviewer suggests as an interesting topic of research is hardly possible to be addressed by ourselves. For such a question, objective measurements of noise and odour annoyance in the environment where individuals live is needed. While such data exists, the level of aggregation and the time of measurement is problematic: in order to match this data to our data we would need precise data on respondents’ locations because data on objective exposure is quantified based on very small spatial units which is not available in our data. In addition, noise exposure data focuses often on one specific source of noise at a time, like roads, train traffic or airports, whereas our study takes into account a general measure of noise annoyance based on different sources of noise annoyance.

Levels of odour are linked to seasonal circumstances (e.g., linked to the seasonal use of fertilisers in agriculture, or to distance to the source of noise, say a highway). A precise measurement of the objective circumstances would require to take into account the position of the individuals and the timing of measurements to the data collection, and in the best case scenario would result in person-centered aggregates. 

In our approach, instead of focusing on the mediating role of perceptions (and use contextual data that is subject to measurement biases as explained above), we preferred to thoroughly control for the objective circumstances by using dummy variables for the area of residence as well as reports on the presence or absence of sources of noise or odour annoyance (like a gas station, wind turbine, airport, busy street, etc). 

It is well possible that noise annoyance and psychological distress are both signs of some personality trait (e.g. general sensitivity to environmental factors, in particular noise sensitivity) that is responsible for both effects. Under that assumption, it is no surprise that both are related to each other. 

However the authors make no effort to discuss such mechanisms as a potential alternative explanation for their findings.

We agree with this point, also raised by reviewer #1. We have now addressed this point in our Conclusions section: 

“Finally, as stated in the introduction, there are many variables that influence both noise and odour annoyance on the one hand and psychological distress on the other hand. We cannot rule out that underlying personality traits (like sensitivity to environmental factors) influence both noise annoyance and noise sensitivity (Van Kamp & Davies, 2013). Future studies should investigate this relationship in more detail.” (page 27)

To cut a long story short, the survey sample lacks the collection of conceptionally important additional variables, namely environmental exposure data on one hand, in the present case, noise exposure from roads, railways, and aircraft (e.g. from so called noise maps), and noise sensitivity data on the other (e.g. measured with the Weinstein scale). This would have opened a more interesting avenue to data analysis, e.g. the possibility to answer the question if noise annoyance is a mediator or moderator variable in the link between noise exposure and psychological distress.

I therefore suggest to reject the manuscript.

We regret to read that the reviewer’s suggestion is to reject our manuscript. We hope that the responses that we provided to the points raised will open a discussion regarding the merits of our work. 

In the following, I list further comments page by page.

COMMENTS BY SECTION:

Page 2, Abstract: "distress and." doesn't make sense.

In the revised manuscript, this mistake is corrected. 

Page 3: Some of the sentences in the text are not very precisely worded in my view. E.g. Page 3, 1st paragraph: a "stressor" cannot be a "reaction" at the same time. 

We have read the manuscript carefully and to the best of our capacities, we have improved the precision of the language. 

One does not "model noise annoyance and odor annoyance" at the same time, but psychological distress (because psychological distress is the dependent variable here).

We have revised this sentence as follows (page 4): 

By taking into account both noise and odour annoyance, this study will provide more insight into their relative contribution to psychological distress (Oiamo, Luginaah, & Baxter, 2015). 

Page 5: The disquisition about the theoretical background is a bit too long for my taste.

We suspect that the reviewer’s comment possibly reflects different customs in writing between social science fields. In our field - sociology, the length of the theoretical background is within the usual limits.

Page 6: I don't understand this sentence: "Secondly, regarding the relative impact of noise and odor annoyance on psychological distress, we expect that for odor annoyance there is more variation in the levels of annoyance in comparison to noise (5)."

We have revised the above mentioned sentence as follows: 

Secondly, regarding the relative impact of noise and odour annoyance on psychological distress, we expect that noise annoyance will have a stronger relative impact. For odour annoyance there is more variation in the levels of annoyance in comparison to noise as a result of seasonal influences (Oiamo et al., 2015). Noise, while depending to some extent on the wind direction, has a more constant presence in the environment, which results in a relatively larger burden for psychological distress compared to odour annoyance. (page 7).

Page 7, 2nd paragraph: Again, too many textbook-style phrases. I would considerably shorten this part.

We have revised the paragraph for brevity, as follows:

First, a lack of personal control, defined as ‘the belief that one can master, control and shape one’s own life’ (Ross, Mirowsky & Pribesh, 2001, p. 572) increases the perceived threat posed by the environmental stressors because individuals feel unable to avoid their negative consequences (Ross et al, 2001; Schafer & Upenieks, 2015). As a result, individuals with low levels of personal control, i.e. women, older adults and less educated (Ross & Mirowsky, 2006, 2009) are unlikely to act to change their situation and they develop negative emotions (Carver, Scheier, & Weintraub, 1989). (page 8)

Page 9, the following doesn't make sense to me: "Additionally, to test the relationship between noise and odor annoyance across more age groups (as we expect older adults’ psychological distress to be more strongly affected by noise and odour annoyance), we use a larger sample which includes also older adults aged 65+ (n = 34838, of which 25326 adults and 9512 older adults). We did not use this sample in our main analysis because these respondents only answered a subset of the items in the noise annoyance scale (three out of nine) and all respondents originated from the same subregion (South-East Brabant)." Why do authors mention this if they don't use that sample?

We perform additional analysis on this sample, therefore we mentioned why this larger sample was not used in our main analysis. For clarity we have revised the text as follows:

“We replicate the main models, on a larger sample which includes older adults aged 65+ (n = 34838, of which 25326 adults and 9512 older adults) because we expect older adults’ psychological distress to be more strongly affected by noise and odour annoyance. This sample was not used in the main analyses because it was drawn from the same subregion (South-East Brabant) and the respondents answered a subset of the items in the noise annoyance scales (three out of nine).” (page 10)

Page 10, first paragraph: Authors should give some more information about the sampling procedures that were applied... (and maybe also some info about the background and the goals of the health monitoring project as a whole).

Following the reviewer’s suggestions, in the revised manuscript we have added additional information:

“We used data from the Health Monitor collected by the Dutch Regional Health Services (GGD), Statistics Netherlands (CBS) and the National Institute for Public Health and the Environment (RIVM), collected between September and December in 2016. The data is collected every four years with the purpose of gathering relevant information about the public health situation in the Netherlands on different spatial levels (e.g. national, regional, local).” (page 9)

“The sampling was done by Statistics Netherlands based on the Municipal Personal Records Database. Respondents were invited to participate in a web survey which included topics such as health, lifestyle, perceived neighbourhood quality and social contacts (for more specific information about the questionnaire: visit the website of the National Institute for Public Health and the Environment (2020)). Respondents were also offered a paper questionnaire as an alternative. In our sample, 74.4% of the adults filled in the web survey and the rest the paper survey.” (page 10)

Page 10, first paragraph: It seems to me important to mention the percentage of deleted records and to discuss any missing data analysis options that were considered. Were the data missing at random? 

The full sample of older adults was 29647, of which a final working sample of 25236 (85,1%) remained. 

In order to reduce the missing cases for noise and odour annoyance, we calculated the scale when respondents had a minimum of 1 item with a valid value. For example for noise annoyance, for people who filled in 8 out of 9 items, we computed the mean out of the 8 items (for n = 409). 

We verified if data was not missing at random based on our sociodemographic variables using Little’s multivariate test. Data was not missing at random based on age or educational level, but it was according to gender. Among women, there were more cases with missing values compared to men. 

We included this information on page 10: 

After deleting the cases with missing values (n = 4411, about 15% of the full sample) on the main variables, the final included sample of adult respondents was 25236. Data on noise annoyance and the two types of odour annoyance was not missing at random based on age or educational level, but it was on gender. Among women, there were more cases with missing values compared to men. 

Why was imputation not considered?

We prefer using the actual scores of individuals because imputation is an estimation method and thus, is dependent on the assumptions and the model used for imputation.

Given the reasonable amount of missing values and the size of our sample, we preferred listwise deletion. 

Btw. the correct sentence here should read. "After deleting the cases with missing values on the main variables, the final number of (analyzable) cases was 25236." -- you cannot 'delete' missing values (as your version of the sentence does), but only cases with missing values. (There are quite a few similar language glitches in the text)

We have read and revised the text carefully. The time that we had available for re-submitting the manuscript did not allow us at this moment to use proofreading services. Provided that the manuscript is accepted for publication, we will ensure that such services are consulted.

Page 11, Noise Annoyance: While there is no "objective" metric for measuring odors, it should have been possible to assign noise exposure values for road, rail, and aircraft noise to each of the respondents, e.g. with the use of national noise maps (that surely must exist for the Netherlands). Authors should explain why they did not include noise exposure data.

We have presented previously the reasons for not including objective measurements of noise and odour exposure. To sum-up the situation, the main issue is that we do not have access to data about the respondent’s address. We only know in which district they live and this spatial unit is too large to use to get a good estimate of their exposure to noise and odour. The exposure can be very different across different places within the district if for example one part of the district is located near a highway, farm or airport, whereas the other part is not. 

Page 11, Odor Annoyance: Authors should mention how the odor question was exactly posed, and how many respondents felt such annoyances at their home. I can imagine that the majority of the sample did not sense any disturbing odors in their living environment, i.e. the place they spend most of their time.

In the revised manuscript, we added more information about how the question about odour annoyance was posed to the data and method section: 

We measured odour annoyance using an item that asked the respondents if they experienced annoyance or hinder during the last twelve months in their home due to: a) roads, b) sewerage, c) fireplaces, d) agriculture, e) industry, f) stables, g) manure, h) animal feed, i) a digester or j) aircraft traffic. Answers ranged from 0 (not annoyed) to 10 (highly annoyed).

For odour annoyance due to agriculture, there were more individuals that did not experience this at all (62%) compared to other type of odour annoyance (45%). 

We also added this information to the data and method section on page 12/13. 

In total, 62% of the respondents indicated that they were not annoyed by odour due to agriculture at home and 45% for odour annoyance due to other sources. This makes sense given that agricultural odour annoyance will be more present in rural areas, whereas odour annoyance due to other sources will be omnipresent (e.g. due to fireplaces, roads, sewerage).

Page 13, first line: something is missing here.

We have revised this sentence:

“Following the study by van Deurzen et al. (2016), we used dummy variables to control for differences in the objective living conditions and exposure to noise and odour between the 306 districts in the sample (size between 42 and 272 respondents).” (page 14)

Page 13: dummies = either 'dummy variable' or 'indicator variable'

We have revised the terminology as suggested by the reviewer.

Page 14 - Analytical strategy: Earlier in the manuscript, authors mention that they have a few "hypotheses". If hypotheses are involved, then the alpha level adopted for the statistical analyses should also be given.

In the revised manuscript, we now report p-values in the table. Although this is not standard practice in our field and the journal does not have strict criteria for reporting significance values, we see the value of increased transparency of the results’ reporting.

Page 15: Reformulate "Turning to the results from our ordinary least squares regression, in Model 1 from Table 2, we present a positive and significant effect of noise annoyance on psychological distress." Rather write "we found a positive and significant effect...".

We have revised this sentence as suggested: 

Turning to the results from our ordinary least squares regression, in Model 1 from Table 2, we found a positive and significant effect of noise annoyance on psychological distress. 

Page 16, Model 3: The analyses presented here all call for a thorough analysis of potential multicollinearity between the main predictors, especially because the authors mention that they are "independent". VIF's (variance inflation factors) should be estimated and included in the results. A large VIF on an independent variable indicates a highly collinear relationship to the other variables that should be considered or adjusted for in some way (e.g. by excluding 'double counting' variables from the model).

Before submitting the previous version of the manuscript, we have performed the suggested analysis but we have not reported the results in the text. We summarize the results here. All VIF-values were below 5. The VIF of noise and odour annoyance due to agriculture or other sources respectively was 1.78, 1.82 and 2.03. There are different guidelines on what is acceptable as a VIF (either above 5 or 10 indicate multicollinearity) (Katrutsa & Strijov, 2017; Midi, Sarkar & Rana, 2010). Therefore, we do not apply extra measures due to multicollinearity for these variables. 

Table 2: Footnote for "Beta" ("Beta" is probably Psychology jargon... and might not be understood by people from other disciplines..). Also explain "R2"

We have added additional explanation for the meaning of the terms “beta” in the analytical strategy, as follows: 

For comparison purposes of the strength of the effect, besides the unstandardized coefficients we also present the standardized ones (beta). 

In addition, in the revised manuscript we use the term ‘Variance explained’ instead of R2.

Table 2 and 3: Indicate p-values in a separate column, there are no reasons why p-values are not given in full.

In the revised manuscript, we included the p-values in the tables. As explained above, this is not common practice in our field.

---

## [Decision Letter · Decision Letter 1]

12 Aug 2021

PONE-D-21-06093R1

The relative effects of self-reported noise and odour annoyance on psychological distress: different effects across sociodemographic groups?

PLOS ONE

Dear Dr. Berkers,

Thank you for submitting your manuscript to PLOS ONE. After careful consideration, we feel that it has merit but does not fully meet PLOS ONE’s publication criteria as it currently stands. Therefore, we invite you to submit a revised version of the manuscript that addresses the points raised during the review process.

We look forward to receiving your revised manuscript.

Kind regards,

Brecht Devleesschauwer

Academic Editor

PLOS ONE

Journal Requirements:

Additional Editor Comments (if provided):

The reviewer comments were correctly addressed, and a number of final, minor revisions were requested. It is indeed important to address the reviewer comments **by making changes to the manuscript**, instead of just replying in the rebuttal letter. Readers of the manuscript might have the same questions as the reviewers, but they will of course not have access to the rebuttal letter.

Reviewers' comments:

Reviewer's Responses to Questions

**Comments to the Author**

1. If the authors have adequately addressed your comments raised in a previous round of review and you feel that this manuscript is now acceptable for publication, you may indicate that here to bypass the “Comments to the Author” section, enter your conflict of interest statement in the “Confidential to Editor” section, and submit your "Accept" recommendation.

Reviewer #2: All comments have been addressed

2. Is the manuscript technically sound, and do the data support the conclusions?

Reviewer #2: Yes

3. Has the statistical analysis been performed appropriately and rigorously? 

Reviewer #2: Yes

4. Have the authors made all data underlying the findings in their manuscript fully available?

Reviewer #2: No

5. Is the manuscript presented in an intelligible fashion and written in standard English?

Reviewer #2: No

6. Review Comments to the Author

Reviewer #2: The manuscript greatly improved over the previous version, and, most importantly, there was quite a lot of additional important information (that was previously lacking) in the author's reply. For the manuscript to be acceptable, I suggest a minor revision which includes a few more of the thoughts from the detailed and well written reply letter, namely (1) on the research rationale, (2) on the importance of odour and noise annoyance's contribution to general health, (3) on the missing values analysis (how was type of missingness determined?), (4) on reasons for not using imputation (in particular, regarding missing items in the K-10)

A few other minor things that could still be ameliorated:

- mention that you did not have (for data protection reasons) the full addresses of the subjects (which would have allowed to model the environmental exposures, still a pity you couldn't)

- "noise exposure" rather than "objective level of noise"

- "stress reactions" rather than "stress"

- get rid of the term "hearing issues", there are virtually no cases of hearing loss due to environmental noise (in contrast to industrial noise).

- get rid of or change the sentence "Gender was measured with a dummy..:" This is nonsense: The variable "Gender" does not need to be transformed to a "dummy" as it only has two levels anyway! In statistical parlance, "dummies / dummy variables" are dichotomous variables (0 or 1) derived from a nominally scaled variable that has more than two levels. Usually for the sake of easier implementation in a statistical model and for a simpler interpretation of the coefficients. As an example, the nominal variable "sleep stage" with 6 different levels, namely W, S1, S2, S3, S4, REM can be broken down into 6 binary variables which can be used as predictors in a statistical model. If a person is e.g. in stage REM, that dummy variable gets the value REM=1, whereas all the others get the value 0. This is the reason behind "dummy". (I anyway prefer the term 'indicator variable' because it indicates the presense or absence of something).

- ad. "dummy" again: Check the sentence beginning with Following the study by van Deurzen... we used ... dummy variables... by using dummy variables"

7. PLOS authors have the option to publish the peer review history of their article (what does this mean?). If published, this will include your full peer review and any attached files.

Reviewer #2: No

---

## [Author Response · Author response to Decision Letter 1]

8 Sep 2021

Revision letter: 

Comments by editor:

1. Journal requirements

We have carefully checked and adjusted our references (using the Plos One Endnote template) and figures (using the recommended site) to meet the journal requirements.

In addition, we are willing to make use of a proofreading service to check and improve the quality of writing if the editor requires us to do so. 

2. Additional Editor Comments (if provided):

The reviewer comments were correctly addressed, and a number of final, minor revisions were requested. It is indeed important to address the reviewer comments by making changes to the manuscript, instead of just replying in the rebuttal letter. Readers of the manuscript might have the same questions as the reviewers, but they will of course not have access to the rebuttal letter.

We are very grateful for all the useful feedback we have received in the previous revision. We are happy to note that the reviewers agree that we have handled the initial comments well and that the manuscript has improved greatly.

In this new version of the paper, we have addressed the reviewer comments and reviewed our revision letter to check if we have left out crucial information for the readers.

Reviewer #2: 

The manuscript greatly improved over the previous version, and, most importantly, there was quite a lot of additional important information (that was previously lacking) in the author's reply. For the manuscript to be acceptable, I suggest a minor revision which includes a few more of the thoughts from the detailed and well written reply letter, namely 

(1) on the research rationale,

We have now added information from the previous revision letter in the text. This includes information on why noise and odour annoyance is important from a policy perspective:

Moreover, noise and odour form the top two environmental complaints for European citizens and therefore is an important issue for policy (D-noses consortium, 2019). For policy makers at a municipality or province, sophisticated exposure data is generally hard to interpret and translate into policy. Therefore, studying the relationship between noise and odour annoyance and psychological distress is important, both from a scientific and policy perspective. (page 4)

Moreover, we have added more information about the impact of daily hassles such as noise and odour annoyance and how they contribute to health inequalities across different groups on page 5:

Life is full of low-impact factors (daily hassles, like noise and odour annoyance) that add up and use the coping resources of individuals and in this way have a substantial impact on individuals (Lazarus & Folkman, 1984). Different sociodemographic groups have access to different coping resources and have different daily living patterns, which influence their capacity to cope with environmental stressors. As a result, certain groups might experience higher levels of psychological distress due to perceived noise and odour annoyance. Answering the question whether different groups are impacted differently by these stressors therefore helps to understand the causes of health inequalities across different groups. 

(2) on the importance of odour and noise annoyance's contribution to general health, 

We have discussed this point in the beginning of the introduction on page 3 (see below): 

Two of these stressors that negatively affect mental health are noise and odour, e.g., the World Health Organization Regional Office for Europe (6) estimates that every year more than 1 million healthy years are lost due to traffic noise only. In addition, odour pollution is associated with many health complaints and decreased quality of life [7]. If we add to this the fact that mental disorders, and especially mood disorders, pose a very high burden on the population of Europe and are expected to become even more prominent for population health in the future [8, 9], it is clear that a better understanding of the relationship between odour and noise and mental health is needed in order to mitigate these noxious pathways.

We ask the reviewer to point out where he/she feels that information about general health could be added. 

(3) on the missing values analysis (how was type of missingness determined?), 

In the new manuscript, we have added more information about the missing value analysis on page 11: 

Using Little’s MCAR test, we found that the missing values in our analysis were not completely missing at random (as the p-value was significant at an alpha of 0.05). 

To evaluate if there were differences in the missing values on psychological distress across the different sociodemographic groups, we used a cross tabulation to calculate the chi-square statistic. There were no significant differences in missing values among men and women, but those with lower levels of education and younger individuals were more likely to have missing values compared to those with higher educational levels and older individuals. 

(4) on reasons for not using imputation (in particular, regarding missing items in the K-10)

In the previous letter, we explained why we chose listwise deletion over multiple imputation. We have now included the reasoning in the text on page 11:

We did not consider imputation methods for our missing values, because this is an estimation method which depends on the assumptions of the model that is estimated. Since we had a reasonable amount of missing values, we chose to rely on listwise deletion. 

A few other minor things that could still be ameliorated:

- mention that you did not have (for data protection reasons) the full addresses of the subjects (which would have allowed to model the environmental exposures, still a pity you couldn't)

We have now mentioned this in the data and methods section on page 15:

Because we do not have access (for data protection reasons) to detailed address data, we cannot model actual noise and odour exposure for respondents. Alternatively, to get an indication of noise and odour exposure around the home, we modelled indicator variables for all districts and self-reported noise and odour annoyance around the home. 

- "noise exposure" rather than "objective level of noise"

We used this terminology in order to make an explicit distinction between noise and odour exposure and the perceived noise and odour annoyance measures. However, we agree that noise exposure also captures this difference and have changed the terminology in the paper.

- "stress reactions" rather than "stress" 

We agree that the term stress reaction is more accurate than stress and have changed the terminology in the paper. 

- get rid of the term "hearing issues", there are virtually no cases of hearing loss due to environmental noise (in contrast to industrial noise).

This comment discusses the sentence below which was used to point out the health risks associated with exposure to noise and odour. We have now included concentration issues instead of hearing issues: 

For the case of noise and odour, potential health risks are an increased probability of developing concentration or sleeping issues (for noise) or irritation on the eyes, nose and throat or respiratory issues (for odour) [27, 38, 39]. (page 6)

- get rid of or change the sentence "Gender was measured with a dummy..:" This is nonsense: The variable "Gender" does not need to be transformed to a "dummy" as it only has two levels anyway! In statistical parlance, "dummies / dummy variables" are dichotomous variables (0 or 1) derived from a nominally scaled variable that has more than two levels. Usually for the sake of easier implementation in a statistical model and for a simpler interpretation of the coefficients. As an example, the nominal variable "sleep stage" with 6 different levels, namely W, S1, S2, S3, S4, REM can be broken down into 6 binary variables which can be used as predictors in a statistical model. If a person is e.g. in stage REM, that dummy variable gets the value REM=1, whereas all the others get the value 0. This is the reason behind "dummy". (I anyway prefer the term 'indicator variable' because it indicates the presense or absence of something).

We understand the reviewers point and therefore changed the sentence into: 

For gender, men formed the reference category. (page 14)

- ad. "dummy" again: Check the sentence beginning with Following the study by van Deurzen... we used ... dummy variables... by using dummy variables"

Conventions in our field are to call it dummy or dummy variable, but we understand that there are differences between scientific fields. We have therefore changed the terminology from dummy variable into indicator variables.

---

## [Editor Report · Decision Letter 2]

20 Sep 2021

The relative effects of self-reported noise and odour annoyance on psychological distress: different effects across sociodemographic groups?

PONE-D-21-06093R2

Dear Dr. Berkers,

We’re pleased to inform you that your manuscript has been judged scientifically suitable for publication and will be formally accepted for publication once it meets all outstanding technical requirements.

Kind regards,

Brecht Devleesschauwer

Academic Editor

PLOS ONE
---

## [Editor Report · Acceptance letter]

23 Sep 2021

PONE-D-21-06093R2 

The relative effects of self-reported noise and odour annoyance on psychological distress: different effects across sociodemographic groups? 

Dear Dr. Berkers:

I'm pleased to inform you that your manuscript has been deemed suitable for publication in PLOS ONE. Congratulations! Your manuscript is now with our production department. 

Kind regards, 

on behalf of

Prof. Dr. Brecht Devleesschauwer 

Academic Editor

PLOS ONE